



# Satellite-based sea ice thickness changes in the Laptev Sea from 2002 to 2017: Comparison to mooring observations

Hans Jakob Belter[1], Thomas Krumpen[1], Stefan Hendricks[1], Jens Hoelemann[2], Markus Janout[2], Robert Ricker[1], and Christian Haas[1]

[1]Sea Ice Physics, Alfred Wegener Institute, Helmholtz Centre for Polar and Marine Research, Am Handelshafen 12, 27570 Bremerhaven, Germany
[2]Physical Oceanography of the Polar Seas, Alfred Wegener Institute, Helmholtz Centre for Polar and Marine Research, Am Handelshafen 12, 27570 Bremerhaven, Germany

**Correspondence:** H. Jakob Belter (jakob.belter@awi.de)

**Abstract.** The gridded sea ice thickness (SIT) climate data record (CDR) produced by the European Space Agency (ESA) Sea Ice Climate Change Initiative Phase 2 (CCI-2) is the longest available, Arctic-wide SIT record covering the period from 2002 to 2017. SIT data is based on radar altimetry measurements of sea ice freeboard from the Environmental Satellite (ENVISAT) and CryoSat-2 (CS2). The CCI-2 SIT has previously been validated with in situ observations from drilling, airborne electromagnetic (EM) measurements and Upward-Looking Sonars (ULS) from multiple ice-covered regions of the Arctic. Here we present the Laptev Sea CCI-2 SIT record from 2002 to 2017 and use newly acquired ULS and upward-looking Acoustic Doppler Current Profiler (ADCP) sea ice draft data (VAL) for validation of the gridded CCI-2 and additional satellite SIT products. The ULS and ADCP time series provide the first long-term satellite SIT validation data set from this important source region of sea ice in the Transpolar Drift. The comparison of VAL sea ice draft data with gridded monthly mean and orbit trajectory CCI-2 data, as well as merged CryoSat-2/SMOS (CS2SMOS) sea ice draft shows that the agreement between the satellite and VAL draft data strongly depends on the thickness of the sampled ice. Rather than providing mean sea ice draft the considered satellite products provide modal sea ice draft in the Laptev Sea. Ice thinner than the modal draft is overestimated, while thicker ice is increasingly underestimated by all satellite products investigated for this study. This tendency of the satellite SIT products to better agree with modal sea ice draft and underestimate thicker ice needs to be considered for all past and future investigations into SIT changes in this important region. The performance of the CCI-2 SIT CDR is considered stable over time, however, observed trends in gridded CCI-2 SIT are strongly influenced by the uncertainties of ENVISAT and CS2 and the comparably short investigation period.



# 1 Introduction

Sea ice is one of the most important indicators for climate change in the Earth's polar regions. Two of the primary parameters that are studied in this context are sea ice concentration (SIC) and sea ice thickness (SIT). While knowledge about SIC is widely available it provides limited insight into overall sea ice changes. A joint evaluation of SIC, SIT and sea ice drift is required for the analysis of sea ice mass balance, volume transports and the overall energy balance (Laxon et al., 2013), which comprehensively explain the complex sea ice system.

While in situ measurements of SIC and SIT are limited in time and space, satellite measurements of both parameters provide the means to assess Arctic-wide changes in the sea ice cover. Satellite remote sensing of SIC started in the 1970s with passive microwave sensors (Parkinson et al., 1999) and was further developed, updated and improved by multiple follow-on missions (Comiso and Nishio, 2008; Cavalieri and l. Parkinson, 2012) until today. While these measurements provide about 40 years of continuous SIC records, SIT satellite records of comparable length are not available. The longest existing SIT data record (from 30 2002 to 2017) was published by the European Space Agency's (ESA) Sea Ice Climate Change Initiative (CCI). The current SIT data record is sufficiently long to achieve the objective of a long-term SIT climate data record (CDR) in the Arctic Ocean and is based on radar altimetry data from the Environmental Satellite (ENVISAT, 2002-2012) and from the CryoSat-2 (CS2) mission that was launched in 2010. SIT remote sensing with radar altimetry relies on retrievals of sea ice freeboard and is therefore an indirect method that is based on certain assumptions and parametrizations that introduce a number of uncertainty factors. These 35 uncertainties can be separated into intrinsic uncertainties that arise from the radar measurements themselves and uncertainties that are induced during the ensuing processing. Processing uncertainties include: the impact of snow radar backscatter and surface roughness on radar ranging and thus the retrieved elevation of the ice surface, the correct discrimination of sea ice and lead surface types with evolving altimeter footprints, the unknown variability of snow mass and snow and sea ice density that go into the conversion of freeboard to thickness (Wingham et al., 2006; Laxon et al., 2013; Ricker et al., 2014).

The CCI Phase 2 (CCI-2) SIT product was validated with observational data from multiple sources (Kern et al., 2018). including, in situ drill holes from a number of North Pole (NP) drift campaigns (Kern et al., 2018), observations from airborne and ground-based electromagnetic (EM) measurements (Haas, 2004; Haas et al., 2009, 2010) and ice draft measurements from Upward-Looking Sonars (ULS)(Hansen et al., 2013; WHOI, 2014; NPI, 2018). However, these measurements are limited to specific regions of the Arctic. While NP drill holes data is limited to the central Arctic, most airborne EM flights took place 45 in the vicinity of Fram Strait, Lincoln Sea and in the Chukchi and Southern Beaufort Sea. ULS measurements were limited to Fram Strait (Hansen et al., 2013) and the Beaufort Sea (WHOI, 2014).

The Russian Shelf Seas are a region where observational data is very limited and which therefore has not been considered for the validation of the CCI-2 SIT CDR. At the same time the Russian Shelf Seas are also regarded to be the most important source regions of Arctic sea ice with the Laptev Sea being the origin of most of the sea ice passing Fram Strait (Rigor et al., 50 2002; Hansen et al., 2013; Itkin and Krumpen, 2017). Recent studies indicate a thinning of Arctic sea ice within the Transpolar Drift (Haas et al., 2008) and in Fram Strait (Krumpen et al., 2019). According to Krumpen et al. (2019) this thinning is a consequence of faster ice transport across the Arctic and leads to more frequent interruptions of the first year ice (FYI) flow



from the Russian Shelves towards the Transpolar Drift. Whether fundamental changes of the sea ice cover in the source regions cause the observed thinning of Fram Strait sea ice, needs to be further investigated.

The available CCI-2 SIT CDR has not yet been fully exploited with respect to variability and trends on the Russian Shelves. This is partly due to the lack of validation data but also because the initial aim of the altimtery missions was to measure fluctuations in perennial SIT (Wingham et al., 2006) which is not prevalent in the FYI-dominated Russian Shelf Seas.

In order to close the observational data gap and validate the CCI-2 SIT CDR in this important region of Arctic sea ice we present a new sonar-based sea ice draft data set from the Laptev Sea. This data set consists of ULS measurements from 2013

to 2015 and upward-looking Acoustic Doppler Current Profiler (ADCP) derived ice draft data that was acquired applying the approach of Belter et al. (2019b). Together with the ADCP-derived ice draft time series the full Laptev Sea validation (VAL) data set covers a period from 2003 to 2016. Since moored sonars are capable of detecting all ice types without a bias towards undeformed ice (Behrendt et al., 2015), this new data set provides comprehensive information about the full thickness range.

The objectives of this study are to examine the ESA CCI-2 SIT CDR and use the new in situ data set to evaluate its

performance in the Laptev Sea. We will analyse the time dependent stability of the CCI-2 SIT CDR in order to see whether potential trends in Laptev Sea SIT are caused by actual changes in SIT in the region or by a change in the ability of the satellites and the ensuing processing steps to characterize the Laptev Sea sea ice cover over time. In this context, stability is defined as the constancy of the mean difference of the CCI-2 SIT CDR to the Laptev Sea observational data. Finally, the case study of the 2013/2014 ULS draft time series from the Taymyr mooring will highlight the findings of the presented comparison of satellite

and sonar-derived sea ice draft time series.

The presented analysis will assist the interpretation and support future algorithm development of altimetry-based SIT CDR. It is an important addition to the existing validation data sets (Kern et al., 2018) and might provide the means to assess regional differences in the performance of the CCI-2 SIT products in the Arctic. For the Laptev Sea region the presented sonar-based data provides better interpretation and more confidence in the ESA CCI-2 SIT products. After all, this unique satellite-derived

SIT record can be an important data set for future investigations into volume transports and will complement previous studies on the changes of the sea ice cover on the Russian Shelves.

## 2   Data and methods

### 2.1   Sonar-based ice draft measurements

The Laptev Sea sea ice draft time series were retrieved by two different approaches to derive ice draft from moored sonars.

The full ice draft time series from upward-looking ADCPs and ULSs (VAL) covers a period from 2003 to 2016 and was taken at water depths between 20 and 60 m. The data set consists of multiple one to two year long sea ice draft time series from a total of nine different locations all over the Laptev Sea (Fig. 1). This inconsistency in the location of the measurements is a considerable limitation for the analysis of sea ice draft variability in this region because we are not sampling a single location over the full period but multiple ones over short periods. Nevertheless, this data set provides important validation data to





analyse the performance of satellite-derived sea ice draft over the Laptev Sea region. The proper validation of the satellite SIT products will then allow the targeted analysis of the long-term changes in SIT in this important region of sea ice formation.

### 2.1.1 Upward-Looking Sonar

ULSs measured from September 2013 to August 2015 at the Taymyr and 1893 stations (Belter et al., 2019a). The Laptev Sea ULSs were of the type Ice Profiling Sonar 5 (420 kHz, manufactured by ASL Environmental Sciences Inc.) and operated with

a single vertical beam (1.8° beamwidth) at a sampling frequency of 1 Hz. Ice draft was inferred from measured values of range (distance between device and ice-water interface) and auxiliary measurements of instrument tilt, pressure and temperature at instrument depth (sampling frequency 1/60 Hz). Final sea ice draft time series with an approximate precision of ± 0.05 m were

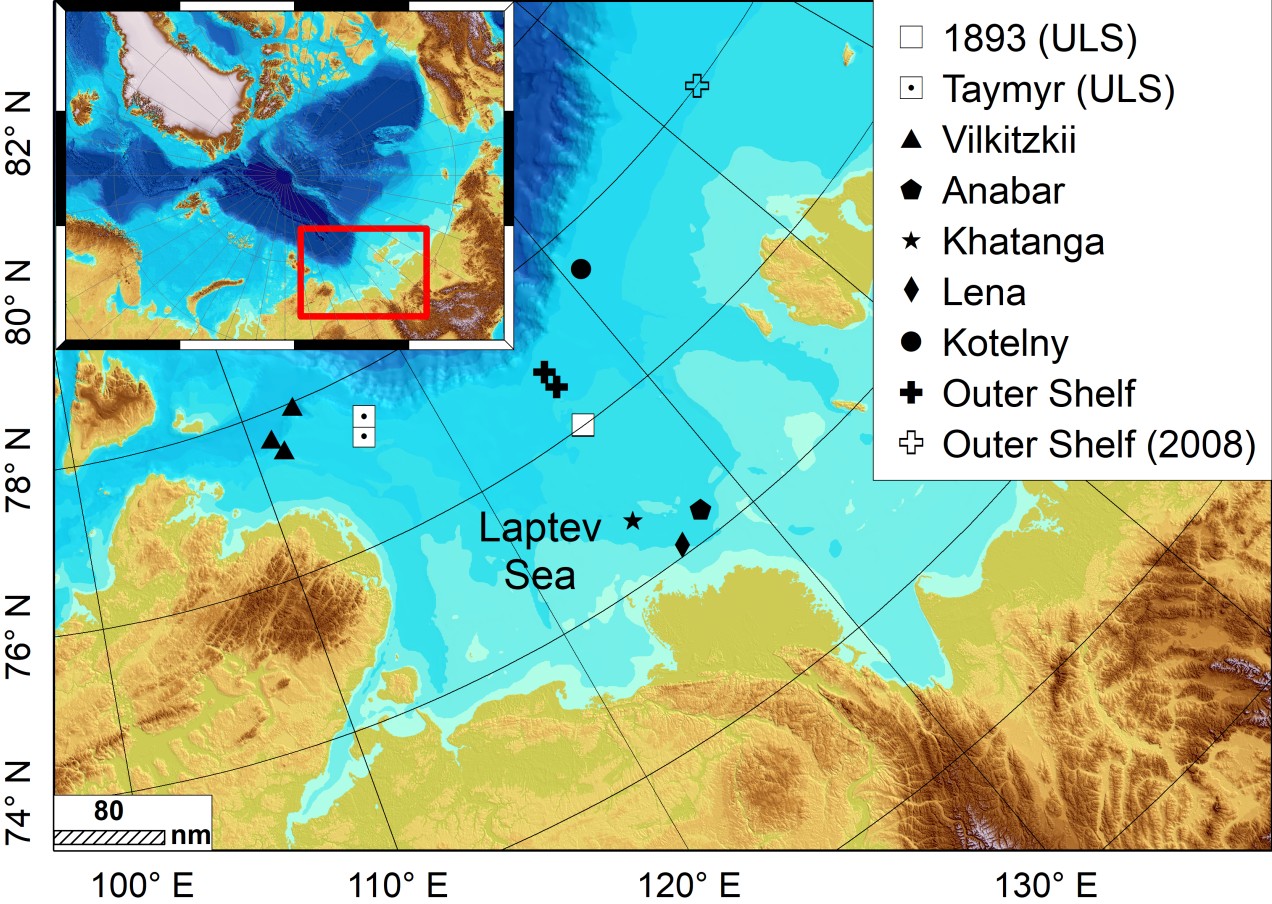

**Figure 1.** Map of the Laptev Sea showing the validation data (VAL) mooring sites. IBCAO basemap provided by Jakobsson et al. (2008).





calculated as the difference between instrument depth and range and corrected for instrument tilts and changes in sound speed (Ross et al., 2016; ASL, 2017).

### 2.1.2 Upward-looking ADCP

The second approach utilized upward-looking ADCPs to derive ice draft time series. Although ADCPs have been used to derive sea ice draft before (Shcherbina et al., 2005; Banks et al., 2006; Hyatt et al., 2008; Bjoerk et al., 2008), the Laptev Sea ADCPs were not equipped with reliable pressure sensors or lacked them altogether. These additional pressure measurements close to the ADCP proved essential for the determination of instrument depth. In order to determine instrument depth without additional pressure data Belter et al. (2019b) proposed an adaptive approach to derive instrument depth using ADCP bottom track (BT) mode measurements of surface and error velocity. Ultimately, their approach yielded daily mean sea ice draft time series with an estimated uncertainty of $\pm\,0.1$ m in the Laptev Sea. Following their method we extended the existing Laptev Sea ULS sea ice draft time series with ADCP-derived sea ice draft in this vastly under-sampled source region of Arctic sea ice. The available ADCPs were upward-looking Workhorse 300 kHz Sentinel ADCPs manufactured by Teledyne RDI. They measured with four different beams (beamwidth 3.8°) at a default angle of 20° from the vertical.

## 2.2 Satellite data

### 2.2.1 ESA CCI-2 monthly mean gridded product

The ESAs CCI-2 SIT Level 3 collated (L3C) gridded product is based on pulse-limited radar altimeter measurements from ENVISAT (2002-2012) and along-track beam-sharpened Synthetic Aperture Interferometric Radar Altimeter measurements from the ongoing CS2 mission (Paul et al., 2018; Hendricks and Ricker, 2019). The CCI-2 SIT data record is available on a $25 \times 25$ km EASE2 monthly grid in the Arctic winter season from October through April. The parameters available from the utilized monthly gridded L3C product include: freeboard, freeboard uncertainty, SIT and SIT uncertainty. For simplicity we distinguish between the CCI-2 ENVISAT gridded data (ENVISAT) for the period from 2003-2012 (Hendricks et al., 2018c) and CCI-2 CS2 gridded data (CS2) for the period from 2010-2016 (Hendricks et al., 2018a). The separation of the two data sets that combine for the full CCI-2 SIT CDR is also required because of the different characteristics of the two satellite radar altimeters. Paul et al. (2018) identified differences in freeboard between ENVISAT and CS2 that are based on waveform parameter variations, footprint differences and the fact that ice surface properties are treated differently. These freeboard differences translate to the gridded monthly mean CCI-2 data presented here. Although Paul et al. (2018) minimized the inter-mission sea ice freeboard biases for the basin average, ENVISAT freeboards in multi-year ice (MYI) regions are still thinner than CS2 freeboards, while ENVISAT provides thicker freeboards than CS2 in regions that are dominated by FYI.

For the comparison to mooring-based VAL sea ice draft data, CCI-2 SIT data was selected from an area of 25 km radius around the VAL mooring location. In order to be consistent with VAL draft data CCI-2 freeboard was subtracted from CCI-2 SIT to obtain gridded monthly mean CCI-2 sea ice draft. Monthly mean CCI-2 sea ice draft uncertainty $d_{unc}$ was calculated as



follows:

$$d_{unc} = \frac{d}{SIT} \cdot SIT_{unc}, \tag{1}$$

where $d$ is sea ice draft and $SIT_{unc}$ is the SIT uncertainty. Finally, all CCI-2 draft data points from within the defined 25 km radius around the mooring site were examined and calculated into a weighted mean sea ice draft value.

### 2.2.2 ESA CCI-2 orbit data

The presented gridded monthly mean CCI-2 data is based on radar altimeter measurements along the orbit trajectories of ENVISAT and CS2 (Hendricks et al., 2018d, b). While the gridded mean data provides Arctic-wide monthly mean values of SIT, the orbital data sets (ENVISATorbit and CS2orbit) provide SIT and freeboard at sensor resolution (2 km in diameter for ENVISATorbit (Connor et al., 2009) and 0.3 km along and 1.5 km across-track for CS2orbit (Wingham et al., 2006)). Due to the frequency of the overflights (approximately four overflights per month for the mooring sites in the Laptev Sea) orbit trajectory data delivers SIT at a higher frequency than the gridded data sets and allows for a comparison of observational data to a larger number of satellite values.

Similarly to the gridded CCI-2 data, CCI-2 orbit data was converted into sea ice draft and draft uncertainty (Eq. (1)). Furthermore, the CCI-2 orbit data was collected from the same 25 km radius around the mooring. In case of the orbit data this was done to account for the area that was sampled by the satellite on the day of the overflight, the ENVISATorbit and CS2orbit data points from inside the 25 km radius were averaged and compared to the corresponding daily mean VAL draft value. The 25 km radius was chosen as it was large enough to ensure a sufficient number of overflights throughout the individual VAL time series. A smaller area limited the available CCI-2 orbit data points, while larger areas included data that was possibly not representative for the VAL data of the overflight day.

### 2.2.3 Merged CryoSat-2/SMOS data

The merged CS2 and Soil Moisture and Ocean Salinity (SMOS) satellite record (CS2SMOS, Ricker et al. (2017)) provides an additional SIT data set with a higher temporal resolution than the gridded monthly mean CCI-2 SIT CDRs. SMOS utilizes 1.4 GHz (L-band) measurements of brightness temperature to retrieve SIT (Tian-Kunze et al., 2014). While the relative uncertainties of the altimetry-based method (CS2) are larger over thin ice regimes (below 1 m thickness), the radiometer-based method (SMOS) shows smaller relative uncertainties over these thin ice regimes (Ricker et al., 2017). Other than gridded CCI-2 and CCI-2 orbit data, CS2SMOS data is only available from 2010 onwards but provides weekly temporal resolution. Furthermore, CS2SMOS combines the advantages of observing thick ($> 1$ m) and thin ($< 1$ m) ice with CS2 and SMOS, respectively, keeping the relative uncertainties for both ice regimes as small as possible (Ricker et al., 2017). Since CS2SMOS is derived by an optimal interpolation of two SIT products and thus does not provide freeboard information, sea ice draft was calculated differently from Eq. (1). CS2SMOS SIT was divided by a constant ratio of 1.136 to compute sea ice draft. This ratio between SIT and draft was derived through nearly 400 drillings of sea ice in Fram Strait (Vinje and Finnekasa, 1986) and is in good agreement with Arctic-wide SIT measurements from Russian drillings (Vinje et al., 1998). In accordance with the CCI-2 data





products, CS2SMOS drafts were also calculated as a weighted mean from all available data points within the 25 km radius around the respective mooring.

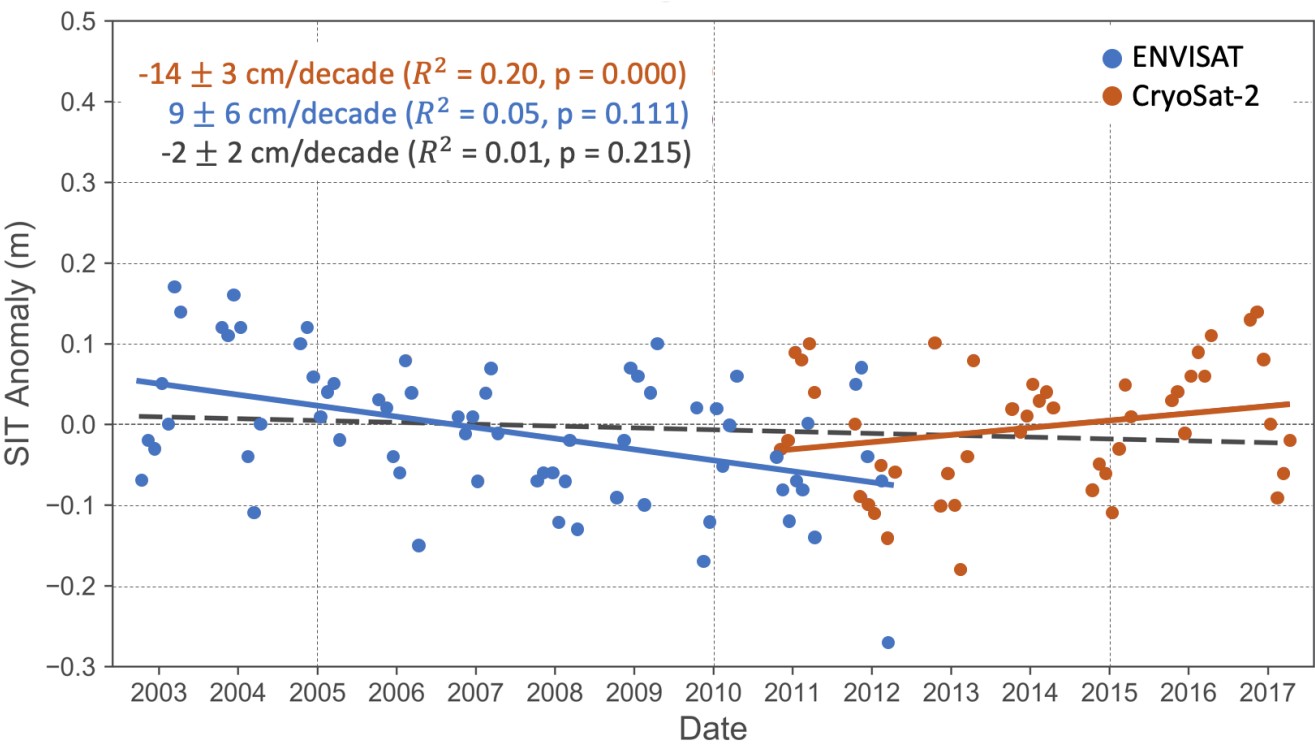

**Figure 2.** ESA CCI-2 gridded sea ice thickness (SIT) anomaly in the Laptev Sea (100-145 °E, 70-81.5 °N). SIT anomaly was calculated for each month compared to the mean of the same month over the full period from 2002 to 2017. Anomalies were calculated for every grid point and averaged over the Laptev Sea area.

# 3 Results

## 3.1 ESA CCI-2 Laptev Sea SIT

The ESA CCI-2 SIT CDR shows an overall thinning of sea ice in the Laptev Sea between 2002 and 2017 (Fig. 2). SIT anomaly was calculated for each month compared to the mean of the same month over the full period from 2002 to 2017. Anomalies were calculated for each grid point and averaged over the Laptev Sea (100-145 °E, 70-81.5 °N). Separating the CCI-2 CDR into the two satellite periods shows that the overall negative trend consists of opposing trends in SIT anomaly from the two CCI-2 data products. While the ENVISAT SIT anomaly (2002-2012) decreases by approximately 14 cm per decade, the trend

in CS2 SIT anomaly shows an increase in SIT from 2010 to 2017. In order to investigate the validity of these satellite-derived trends in SIT anomaly the following section provides the results of the statistical analysis of the differences between VAL and





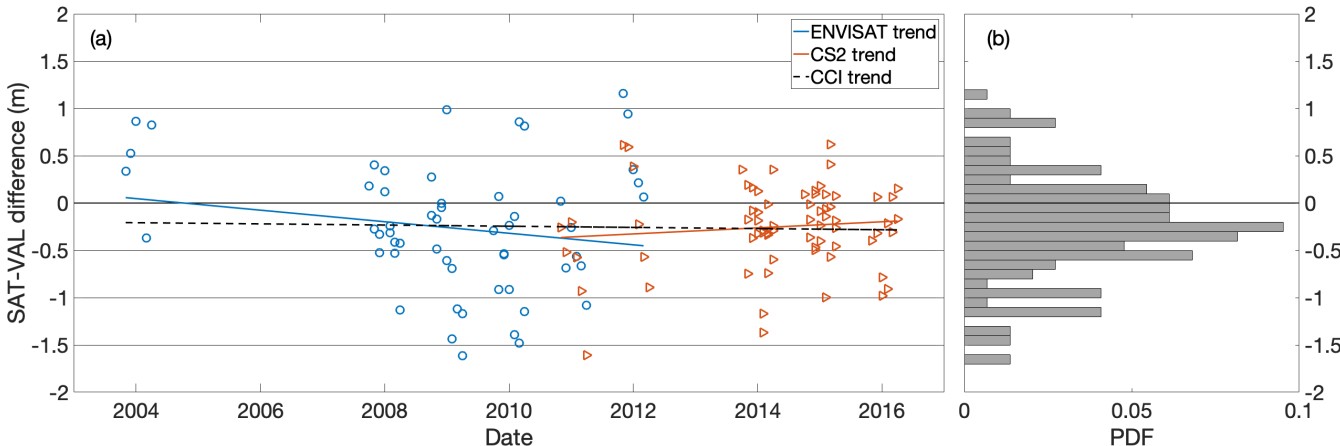

**Figure 3.** (a) Difference (SAT-VAL difference) between gridded monthly mean ENVISAT/CS2 and VAL ice drafts. VAL data consists of ice draft data derived from upward-looking ADCPs for the ENVISAT period (blue) and a combination of ADCP and ULS data for the CS2 period (orange). (b) Probability density function (PDF) of SAT-VAL differences over the full period from 2003 to 2016.

satellite-derived sea ice draft data from the Laptev Sea. To determine the agreement between satellite and VAL sea ice draft data, values of root mean square difference (RMSD), mean difference and correlation coefficient ($r$) were calculated for each of the individual data sets from the stations shown in Fig. 1. For comparison between the ENVISAT and CS2 missions, averages of these three statistical parameters were calculated for all stations during the overlap period from November 2010 to March 2012.

### 3.2 Gridded CCI-2 sea ice draft

Figure 3 shows the differences between gridded monthly mean CCI-2 and VAL sea ice draft (SAT-VAL difference) for the period from 2003 to 2016. Individual SAT-VAL differences show substantial scatter around zero but the overall trend (black line) indicates an almost constant mean difference of approximately -0.3 m over the full investigation period. Table 1 and 2 provide RMSD, mean difference and correlation coefficients between the gridded ENVISAT and CS2 and VAL draft data from each station, respectively.

For the ENVISAT period RMSD values average 0.70 m, with minimum RMSD of 0.37 m for the Anabar 2007/2008 and maximum RMSD of 1.0 m for the Khatanga 2008/2009 data. The average mean difference is -0.22 m indicating an average underestimation of monthly mean sea ice draft by the ENVISAT data. The ENVISAT underestimation of sea ice draft occurs for all but two data sets. Lena 2003/2004 and Outer Shelf 2011/2012 mean differences are 0.44 and 0.55 m, respectively, indicating a mean overestimation of sea ice draft by the ENVISAT product at these stations. The average correlation coefficient between gridded monthly mean ENVISAT and VAL sea ice draft data is 0.44 for the period from 2003 to 2012. Results from multiple




stations show little or almost no correlation, while correlations are significant at the 95% confidence level for data from only

three stations.

Compared to ENVISAT, differences between gridded monthly mean CS2 and VAL sea ice draft show a smaller average RMSD (0.48 m) and a higher mean correlation coefficient (0.50). The average mean difference of -0.27 m is slightly more negative than for ENVISAT. This indicates a stronger mean underestimation of VAL sea ice draft by CS2 compared to ENVISAT. Mean differences are negative for all stations, showing consistent underestimation by CS2 data. Although the mean correlation

coefficient is larger compared to the ENVISAT period none of the individual coefficients is significant at the 95% confidence level during the CS2 period.

By grouping VAL sea ice draft values in 0.2 m bins and comparing them to their corresponding monthly mean ENVISAT (2003 to 2012) and CS2 (2010 to 2016) sea ice draft values we are able to examine the agreement between gridded CCI-2 and VAL drafts along the full range of sea ice drafts that were measured by the moorings (Fig. 4). Both scatter plots indicate an

overestimation by the gridded CCI-2 products for draft values below approximately 0.7 m. The magnitude of the overestimation decreases with increasing draft. The best agreement occurs for draft values between 0.7 and about 1.2 m, while monthly mean VAL sea ice draft is underestimated for draft values above approximately 1.3 m. The underestimation increases with increasing ice draft values. Additionally, Fig. 4 shows that the variability of the ENVISAT draft values is substantially larger within the selected 0.2 m bins compared to CS2 draft values in the same bins. The difference in the performance of ENVISAT and CS2

data is also revealed for the overlap period between the two satellite missions (2010-2012). While mean differences show the same tendency with -0.54 m (ENVISAT, Table 1) and -0.68 m (CS2, Table 2) for the 2010-2011 Outer Shelf data sets, they disagree considerably for the 2011-2012 period (ENVISAT: 0.55 m, CS2: -0.02 m).

## 3.3 Higher temporal resolution satellite products

In order to complement the results shown for the comparison between gridded CCI-2 and mean VAL data, we conducted an

additional analysis with satellite data products that are based on the measurements from the ENVISAT and CS2 missions and the gridded CS2 data but provide higher temporal resolution of sea ice draft than the gridded CCI-2 record. RMSD, mean difference and correlation coefficients were calculated for the comparison of sea ice draft from ENVISATorbit (Table 1) and CS2orbit (Table 2) trajectory data and merged CS2SMOS (Table 3) data with VAL sea ice draft data.

### 3.3.1 Orbit CCI-2 sea ice draft

While the average RMSD, mean difference and correlation coefficients are very similar for the VAL data comparison to gridded CS2 and CS2orbit, almost all stations show significant (at the 95% confidence level) correlations between CS2orbit and VAL sea ice draft (Table 2). ENVISATorbit data shows a higher average RMSD, stronger average underestimation of VAL sea ice draft and much lower average correlation with VAL sea ice drafts compared to the gridded ENVISAT data (Table 1). This suggests that the CS2 component of the CCI-2 CDR is superior to the ENVISAT sea ice draft data. It also confirms the inter-

mission biases between ENVISAT and CS2 that were published by Paul et al. (2018).





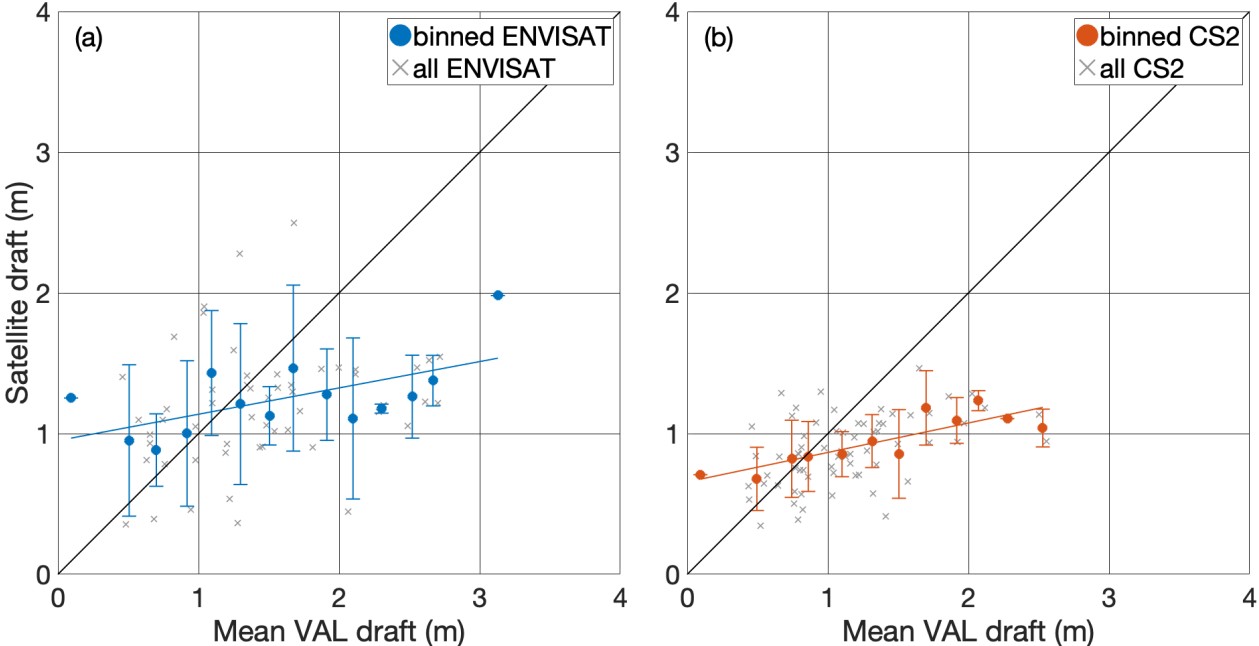

**Figure 4.** Scatterplot comparing gridded monthly mean CCI-2 sea ice draft to VAL sea ice draft (grey crosses). Panel (a) shows the comparison for the ENVISAT period superimposed by the mean ENVISAT draft per 0.2 m VAL data bin, while panel (b) shows the same for CS2 data and period. Error bars indicate ± one standard deviation of the CCI-2 data within the specific 0.2 m bin.

### 3.3.2 Merged CS2SMOS sea ice draft

The comparison of weekly CS2SMOS and VAL sea ice draft data reveals the largest average correlation coefficient. On the other hand, the CS2SMOS and VAL draft comparison also shows the largest average underestimation of any of the presented satellite data products. Figure 5 shows the comparison of the agreement between the gridded and orbit CCI-2 and the CS2SMOS

data products with the corresponding VAL sea ice draft data. While the overall tendency of the gridded CCI-2 products to overestimate ice draft for thin ice and increasingly underestimate thickening ice is confirmed by CCI-2 orbit and CS2SMOS data a general offset between the individual satellite products is visible for most of the selected 0.2 m VAL data bins. While both ENVISAT draft data sets indicate thickest drafts over the full thickness range, gridded CS2 and CS2orbit agree rather well. CS2SMOS data shows smallest draft values throughout the entire thickness range compared to the CCI-2 products.

The overestimation of sea ice draft values below 0.7 m that is apparent in the gridded and orbit CCI-2 data is minimized by the impact of SMOS on the merged CS2SMOS product. The Laptev Sea is dominated by newly formed and thinner FYI, accordingly the gridded merged product is dominated by SMOS data. Consequently the underestimation of sea ice draft with increasing thickness is largest for CS2SMOS because of the larger uncertainties of SMOS over thicker sea ice.


In summary, the gridded CCI-2 products underestimate monthly mean sea ice draft in the Laptev Sea by an average of -0.22 m

(-0.27 m) during the ENVISAT (CS2) period. While individual stations deviate from this average the overall tendency indicates a thickness dependency of the agreement between monthly mean gridded CCI-2 and VAL sea ice draft. Thin ice (<0.7 m) is overestimated by the gridded CCI-2 products and thicker ice (>1.3 m) is increasingly underestimated with increasing ice draft. The overall spread in SAT-VAL difference values is smaller for the CS2 period. ENVISATorbit and CS2orbit and merged CS2SMOS sea ice draft data, which provide higher temporal resolution than the gridded monthly mean products confirm these

results. However, it has to be noted that sea ice draft values from the four presented satellite products deviate considerably from one another.

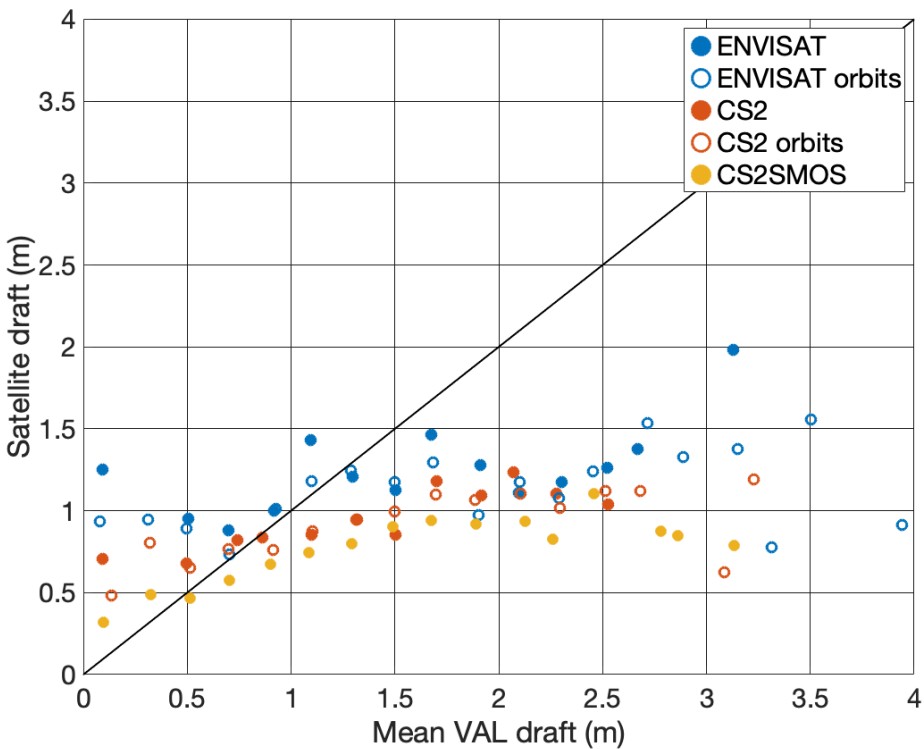

**Figure 5.** Mean sea ice drafts per 0.2 m VAL data bin from ENVISAT (filled blue circles), ENVISATorbit (blue circles), CS2 (filled orange circles), CS2orbit (orange circles) and CS2SMOS (filled yellow circles) data products.





## 4   Discussion

### 4.1   Comparability of satellite and sonar measurements

ENVISAT and CS2 average mean differences to VAL sea ice draft are of similar magnitude, which indicates a consistent
average underestimation of Laptev Sea sea ice draft from the gridded monthly mean CCI-2 CDR between 2003 and 2016.
In order to discuss these results and most importantly their meaning for the apparent trends in CCI-2 SIT in the Laptev Sea
(Fig. 2) the deficiencies of the VAL and CCI-2 data products have to be examined.

The comparison between point measurements from moorings and gridded satellite products is by default challenging. A
significant difference between sonar and altimetry-based measurements are the parameters that are measured. While moored
sonars provide sea ice draft data, radar altimeters infer SIT from measurements of freeboard. Altimeter freeboard is converted
into SIT based on parametrizations of snow depth and constant densities of snow and sea ice. Snow depth, snow and sea
ice density are parameters that are not routinely measured and therefore are based on climatologies: modified Warren snow
climatology and Warren snow water equivalent climatology (Warren et al., 1999; Ricker et al., 2014). These assumptions
contribute to the uncertainties of the final SIT data records and consequently to the CCI-2 sea ice draft values that are calculated
for the presented comparison to VAL sea ice draft. Additionally, both measurements take place on completely different spatial
scales. Moored sonars sample a single point throughout the respective sampling period. In contrast, the location of radar
altimetry measurements is defined by footprints of the instruments and the trajectories of the satellites. Additionally, the final
CCI-2 data product is gridded to achieve Arctic-wide coverage which means that variability within an $25 \times 25$ km grid cell
is not resolved. These fundamental differences between the compared measurement principles have to be considered when
comparing the presented satellite and sonar-based sea ice draft data sets. Additionally, VAL and CCI-2 time series are derived
from multiple different instruments during the investigated period from 2003 to 2016. Accordingly, each of these individual
records consists of data from different measurement configurations themselves.

### 4.2   Data limitations

#### 4.2.1   VAL data

VAL data is based on sonar-derived ice drafts from two differing instruments. In general, the default setup, with a single
narrow vertical beam and a sampling frequency of 1 Hz, makes the ULS the primary instrument for stationary long-term
observation of sea ice draft. Although upward-looking ADCPs are based on the the same measurement principles they are
build for measurements of currents and ice drift rather than sea ice draft. Consequently, the ADCP-derived sea ice draft time
series are less accurate than ULS-derived time series (Belter et al., 2019b). As a result this study compares satellite data to
VAL data sets of different quality. This compromise in data quality between ULS and ADCP was taken on because we consider
the uncertainty of approximately 0.1 m (Belter et al., 2019b) of the ADCP-derived daily mean sea ice draft time series to be
sufficiently accurate for the comparison to weekly and monthly mean sea ice draft from gridded satellite products. Since they
are of sufficient quality, the ADCP-derived draft records allow us to significantly extend the available ULS-derived time series.



Rather than analysing data from only two consecutive years we are able to investigate a time period of almost 13 years. The
increased length of this unique Laptev Sea VAL data set is vital for the evaluation of the stability of the investigated CCI-2
records.

Despite the fact that we were able to extend our Laptev Sea VAL data set it has to be noted that in situ observations of sea
ice draft are very limited in the Laptev Sea. The lack of mooring measurements over more than two years at any of the sampled
locations prohibits us from comparing satellite data to VAL data from a single mooring location. Instead, the entire VAL data
record is composed of one to two year time series from a total of nine different locations all over the Laptev Sea (Fig. 1).
Although this inconsistency is unfavourable for the analysis of long-term variability of sonar-based SIT in this region the VAL
data provides a new and unique validation record for the CCI-2 SIT CDR.

### 4.2.2 ESA CCI-2 gridded monthly mean draft data

Like the VAL data record, gridded and orbit CCI-2 data is based on measurements from two different systems. Inter-mission
differences have been analysed previously and indicate that due to the different setups of the ENVISAT and CS2 radar altimeters
the final SIT, and therefore draft, records contain residual intermission differences (Guerreiro et al., 2017; Paul et al., 2018).
These biases vary regionally and seasonally. The seasonal biases between ENVISAT and CS2 need to be considered for the
temporal development of the Laptev Sea SAT-VAL differences between the two periods. For the Laptev Sea the ENVISAT SIT
is, on average, approximately 0.22 m thicker than the CS2 SIT for the overlap period from November 2010 to March 2012
(Paul et al., 2018).

In addition, the biggest limitation for the analysis of the performance of the gridded CCI-2 CDR is its temporal resolution
of one month and its limitation to the period from October through April. This significantly limits the number of CCI-2 draft
data points for the comparably short validation period from 2003 to 2016.

### 4.3 Stability of the CCI-2 SIT CDR

In general, the stability of the satellite records is defined as the constancy of the SAT-VAL differences over time. However, the
fact that the full VAL data record consists of multiple one to two year sea ice draft time series from various stations all over the
Laptev Sea rather than a single time series from one location inhibits us from assessing an overall trend in sea ice draft over the
full VAL period. Therefore, the observed near-consistent average mean differences over the ENVISAT and CS2 periods (Fig. 3)
do not provide enough proof of a stable performance of the gridded CCI-2 data. SAT-VAL differences are dependent on the
thickness of the ice that is sampled, which means in order to investigate the stability of the gridded CCI-2 records, SAT-VAL
differences need to be analysed for different thickness ranges. We therefore consider the presented gridded CCI-2 draft record
stable only if the SAT-VAL differences within the selected thickness ranges stay constant over time.

The limiting factor for the analysis of temporal changes in the SAT-VAL difference from different thickness ranges is, again,
the small number of data points and the comparably short observational period. The following thickness ranges were selected
in order to provide a reasonable number of data points for the analysis of trends: 0 to 1 m, 1 to 2 m and 2 to 3 m (Fig. 6). For the
thickness ranges between 0 and 1 m and 1 and 2 m negative trends are visible while a positive trend is apparent for the thickness




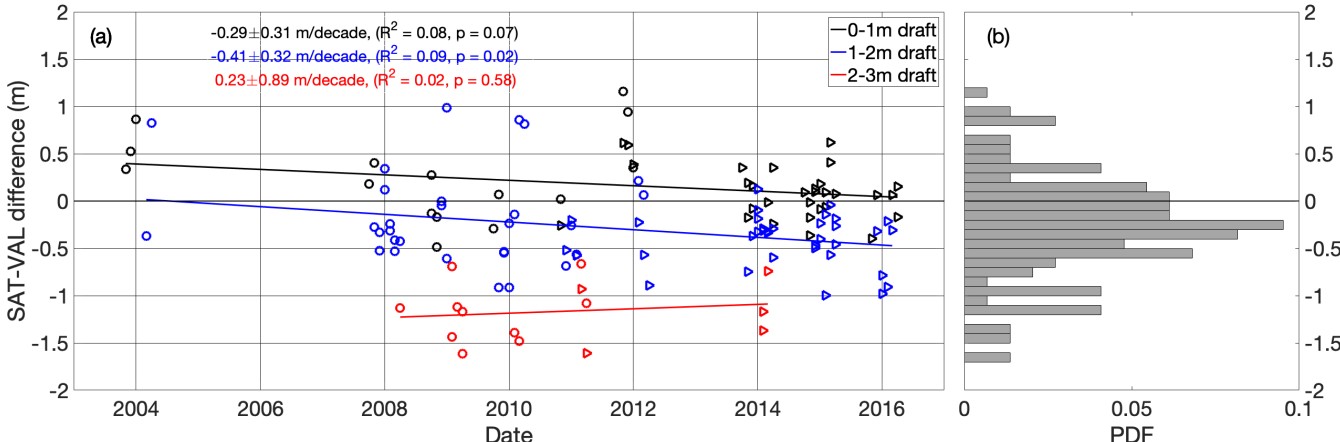

**Figure 6.** (a) Difference (SAT-VAL difference) between gridded monthly mean ENVISAT (CS2) and VAL ice drafts in circles (triangles) for thickness ranges from 0 to 1 m (black), 1 to 2 m (blue) and 2 to 3 m (red). Linear trends were computed for each of the thickness ranges. (b) Probability density function (PDF) of SAT-VAL differences from 2003 to 2016.

range from 2 to 3 m. However, the coefficients of determination, $R^2$, for all three trends are very small indicating that linear trends poorly represent the Laptev Sea SAT-VAL difference and are in fact not suitable to explain the temporal development of SAT-VAL differences over time. Nevertheless they allow us to investigate the stability of the mean difference for different

thickness ranges. The trends indicate a decrease (increase) in mean difference for the thickness ranges 0 to 1 m and 1 to 2 m (2 to 3 m). All three trends have large uncertainties and only one is significant at the 95% confidence level (1 to 2 m thickness range, $p$-values below 0.05). These trends are dependent on the length of the observed time series and in the presented case on the inter-mission biases between the two CCI-2 products that combine for the full gridded CCI-2 sea ice draft CDR. The above-mentioned ENVISAT overestimation of freeboard in FYI-dominated regions like the Laptev Sea leads to an overestimation of

ice draft compared to CS2. SAT-VAL differences during the overlap period (2010 to 2012) show larger differences between satellite and VAL draft for ENVISAT than for CS2 (Fig. 3). This tendency of the ENVISAT data to generally provide thicker ice in FYI regions than CS2 can also be seen in Fig. 4 and might explain the negative trends observed in the 0 to 1 and 1 to 2 m thickness ranges (Fig. 6). The underestimation of thicker ice by ENVISAT compared to CS2 (Paul et al., 2018) on the other hand is visible in the positive trend in SAT-VAL difference that is indicated for the 2 to 3 m thickness range.

Based on this analysis we consider the trends within the three thickness ranges to be caused by the inter-mission bias between ENVISAT and CS2 and the overall gridded CCI-2 CDR to be stable for the investigated period from 2003 to 2016.

### 4.4 Taymyr 2013/2014 case

In order to support the interpretation and underline the current deficiencies of satellite-derived sea ice draft data in the Laptev Sea we present a case study based on the 2013/2014 ULS deployment at Taymyr station (Fig. 7).





The Taymyr station is located in the western Laptev Sea (Fig. 1). The region is dominated by offshore winds that open coastal polynyas. The ice formed in these polynyas is transported northwards (Itkin and Krumpen, 2017) and passes by the mooring site. Changes in wind direction can lead to temporary closing of the polynyas and convergence towards the coast or fast ice. Sea ice piling up against the south-western coast is deformed and increases in thickness.

    We utilized a Lagrangian tracking tool, ICETrack (Krumpen, 2017), to determine the trajectories of the ice that was passing
by the mooring. The tracking provides us with information about the source regions of the ice measured by the ULS and the atmospheric and oceanic conditions the ice experienced on its trajectory to the mooring location. The NSIDCs Polar Pathfinder sea ice motion product (Tschudi et al., 2019) was used to estimate convergence along the trajectories of the Taymyr sea ice. Analysing daily convergence along the trajectories allowed us to calculate accumulated convergences over each track. Accumulated convergence is a measure for the total amount of deformation the ice that passed by the Taymyr mooring has
experienced before it reached the mooring site.

    The daily mean ULS draft time series from the Taymyr station indicates a consistent increase in sea ice draft between January and March 2014. Since the Laptev Sea is dominated by newly formed FYI ice the observed daily mean draft values cannot be explained by thermodynamic growth only. An additional dynamic influence on the ice is confirmed by the increase in accumulated convergence along the trajectories over the same period from January to March 2014. When comparing the

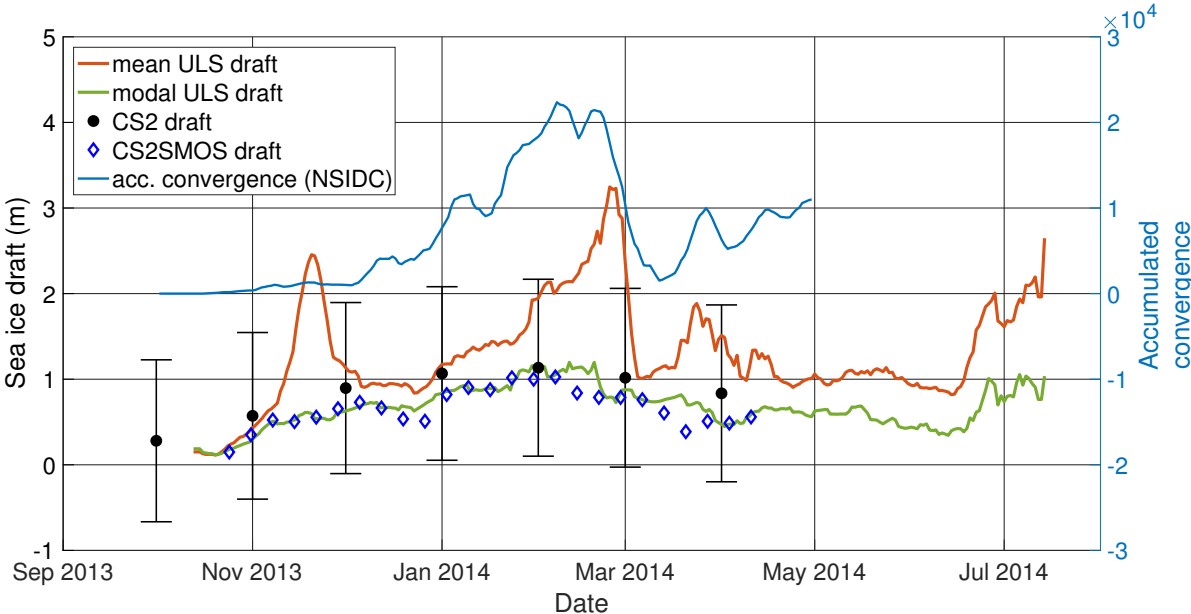

**Figure 7.** Time series of CS2 (black circles) and CS2SMOS (blue diamonds) sea ice draft compared to ULS-derived mean (orange) and modal (green) sea ice draft from Taymyr station (2013-2014). Black error bars indicate uncertainty of the monthly mean gridded CS2 draft. Sea ice passing the mooring site was tracked using the Lagrangian ice tracking tool ICETrack (Krumpen, 2017). Based on the NSIDC Polar Pathfinder sea ice motion product (Tschudi et al., 2019) accumulated convergence (blue) along the daily sea ice trajectories was calculated.



daily mean ULS time series to the gridded monthly mean CS2 draft time series it is apparent that the CCI-2 product is not
able to reproduce the dynamic increase in sea ice draft. Rather than showing the mean sea ice draft CS2 data shows better
agreement with the modal sea ice draft derived from the ULS (Fig. 7). A similar result is visible for the weekly draft values
from CS2SMOS. Table 4 shows RMSD, mean difference and correlation coefficients for the comparison between gridded CS2,
CS2orbit and CS2SMOS with modal sea ice draft data from the ULS moorings (Taymyr and 1893) for the period from 2013

to 2015. Gridded CS2, CS2orbit and CS2SMOS show small mean differences to modal sea ice drafts in the Laptev Sea. Mean
correlation coefficients between modal ULS and mean satellite data are between 0.61 and and 0.77 and significant at the 95%
confidence level for the higher temporal resolution satellite products (CS2orbit and CS2SMOS).

Another important observation from this case study concerns the differences in length between the satellite and VAL sea ice
draft time series. While satellite data is only available from October and through April, VAL time series indicate that sea ice

persists until summer (July in this case). It is known that warm snow and ice as well as the formation of melt ponds prevent
CS2 retrieval of Arctic SIT between May and September (Ricker et al., 2017). That means that, for investigations into the sea
ice cover in the Laptev Sea it is important to be aware that sea ice can persist some time after the presented satellites stop
providing SIT data.

## 5  Conclusions

The ESAs CCI-2 gridded SIT CDR covers a period from 2002 to 2017 and has been validated for multiple regions around
the Polar regions of the Earth. The presented in situ observations of sea ice draft from Laptev Sea ULS and ADCP moorings
provide an additional important validation data set from one of the most under-sampled FYI-dominated regions of Arctic sea
ice.

The comparisons between sea ice draft data from ULS and upward-looking ADCPs with gridded monthly mean CCI-2 sea
ice draft, higher resolution CCI-2 orbit trajectory and the merged CS2SMOS data in the Laptev Sea indicate:

- The agreement between in situ sonar and satellite data is very sensitive to the thickness of the sampled sea ice.

- Sea ice drafts below 0.7 m are overestimated, while sea ice drafts above approximately 1.3 m are increasingly underestimated by all considered satellite data products.

- The presented satellite products represent similar sea ice drafts differently.

The Taymyr 2013/2014 case study highlights the current deficiencies of the satellite-derived SIT records in the FYI-dominated
Laptev Sea region:

- Rather than representing mean sea ice draft, the considered satellite products show better agreement with modal sea ice draft.

- Significant, lasting deformation events that lead to large mean sea ice drafts are not represented in any of the shown
satellite data products.





The presented stability analysis of SAT-VAL draft differences reveals that the agreement between gridded monthly mean CCI-2 and VAL sea ice draft data is dependent on the thickness of the ice that is sampled but mean differences are consistent over time for similar thicknesses. Linear changes in mean differences for individual thickness ranges are attributed to inter-mission bias in SIT representation between the two missions (ENVISAT and CS2) composing the gridded CCI-2 record and the comparably
small number of data points that were available for the individual thickness ranges.

Applying these results to the presented Laptev Sea CCI-2 SIT anomaly trends (Fig. 2) we conclude that the trends of the ENVISAT and CS2 component are not caused by a change in the performance of the CCI-2 products over time but rather actual changes in SIT in this region. However, due to the high uncertainties of the data products and the comparably short sampling periods these trends need to be investigated further. Although, the stability analysis provides confidence in the CCI-2
SIT CDRs it has to be noted that satellite-derived SIT data is not sufficient to explain overall changes in SIT in the Laptev Sea. In agreement with Haas (2004) we conclude that current satellite SIT data allows examination of changes in modal SIT and therefore the thermodynamic component of the changes in the Laptev Sea, however, dynamic changes in SIT are not reproduced by the satellite CDRs. Therefore, improvements in the processing of radar altimetry data are required for the estimation of surface roughness but also for the parametrizations of snow depth and densities of snow and ice. Unknown snow
depth distribution is a major source for uncertainty in the freeboard retrieval process. Uncertainties in freeboard as well as slight changes in the utilized average ice column densities translate into the final SIT product. As suggested by Wingham et al. (2006) ice type densities should be replaced by thickness dependent ice densities to account for the currently unknown density variations due to deformation processes. Furthermore would continuous long-term SIT measurements in the Laptev Sea provide much needed information on deformation processes. However, with limited access to the vastly under-sampled Russian
Shelf regions the satellite-derived SIT CDRs remain a crucial source of long-term SIT data for this region. Their improvement as well as large-scale observations of dynamic changes of SIT redistribution and model simulations are required to investigate the effects governing SIT changes in the Laptev Sea.

*Data availability.* ULS (Belter et al., 2019a) and ADCP-derived daily mean sea ice draft time series (data in the process of being published) are available at the Data Publisher for Earth & Environmental Science PANGAEA.
ESA Sea Ice Climate Change Initiative (Sea_ Ice_ cci): Northern hemisphere sea ice thickness from ENVISAT satellite (Hendricks et al., 2018c) and from CryoSat-2 satellite (Hendricks et al., 2018a) on a monthly grid (L3C), v2.0 are available from the Centre for Environmental Data Analysis data base.

ESA Sea Ice Climate Change Initiative (Sea_ Ice _ cci): Northern hemisphere sea ice thickness from ENVISAT (Hendricks et al., 2018d) and CryoSat-2 (Hendricks et al., 2018b) on satellite swath (L2P), v2.0 are available from the Centre for Environmental Data Analysis data
base.

Merged CryoSat-2/SMOS sea ice thickness: A weekly Arctic sea-ice thickness data record from merged CryoSat-2 and SMOS satellite data (2017). The Cryosphere, 11, 1607-1623, https://doi.org/10.5194/tc-11-1607-2017





*Author contributions.* HJB carried out the analysis, processed ULS and ADCP data and wrote the manuscript. All authors contributed to the discussion and provided input during the writing process. In addition to their input to the manuscript TK conducted the backward-tracking
of sea ice from the individual mooring sites, SH provided CCI-2 data and conducted the analysis of CCI-2 SIT changes in the Laptev Sea and RR contributed CS2SMOS SIT data to the analysis. JAH and MAJ deployed and recovered the moorings during numerous expeditions to the Laptev Sea.

*Competing interests.* Author CH is a member of the editorial board of the The Cryosphere. All other authors declare that they have no conflict of interest.

*Acknowledgements.* This study was carried out as part of the BMBF-funded Russian-German research cooperation QUARCCS (grant: 03F0777A). Moorings were deployed and recovered within the framework of the Russian-German project CATS/Transdrift (grant: 63A0028B). Special thanks to all the people involved on the various expeditions.

The 2013/2014 ULS data sets were processed by ASL Environmental Sciences Inc., Victoria, BC, Canada. ASL also provided valuable support and the toolboxes for the processing of the 2014/2015 ULS data sets. Additionally, the ECMWF provided ERA-Interim reanalysis
surface pressure data (Dee et al., 2011) that was valuable for the ULS processing.

The production of the merged CryoSat-SMOS sea ice thickness data was funded by the ESA project SMOS & CryoSat-2 Sea Ice Data Product Processing and Dissemination Service, and data from 2010 to 2016 were obtained from AWI.



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





**Table 1.** Statistics of the comparison of gridded monthly mean ENVISAT and ENVISATorbit draft data with VAL mean sea ice draft for the period from 2003 to 2012. RMSD and mean difference were calculated for the differences of ENVISAT minus VAL mean sea ice draft. The Pearson correlation coefficient, *r*, was calculated for each station. Bold correlation coefficient values indicate significant correlation at the 95% confidence level. Bottom line values show the averages of RMSD, mean difference and correlation coefficient over all stations.

| Period | Station | ENVISAT | | | ENVISATorbits | | |
|---|---|---|---|---|---|---|---|
| | | RMSD | Mean difference | r | RMSD | Mean difference | r |
| | | [m] | [m] | | [m] | [m] | |
| 2003-2004 | Lena | 0.63 | 0.44 | 0.25 | 0.95 | 0.02 | -0.05 |
| 2007-2008 | Anabar | 0.37 | -0.17 | 0.53 | 0.75 | -0.30 | -0.01 |
| | Khatanga | 0.54 | -0.30 | 0.43 | 1.20 | -0.60 | -0.01 |
| 2008-2009 | Khatanga | 1.00 | -0.45 | -0.14 | 1.06 | -0.61 | -0.02 |
| | Outer Shelf | 0.73 | -0.60 | **0.90** | 0.92 | -0.65 | **0.54** |
| 2009-2010 | Anabar | 0.75 | -0.14 | 0.05 | 0.84 | -0.09 | 0.20 |
| | Khatanga | 0.92 | -0.72 | **0.81** | 1.11 | -0.73 | 0.11 |
| 2010-2011 | Outer Shelf | 0.64 | -0.54 | **0.86** | 0.84 | -0.61 | **0.60** |
| 2011-2012 | Outer Shelf | 0.69 | 0.55 | 0.29 | 0.65 | 0.27 | 0.12 |
| **2003-2012** | **Mean** | 0.70 | -0.22 | 0.44 | 0.93 | -0.37 | 0.16 |





**Table 2.** Statistics of the comparison of gridded monthly mean CS2 and CS2orbit draft data with VAL mean sea ice draft for the period from 2010 to 2016. RMSD and mean difference were calculated for the differences of CS2 minus VAL mean sea ice draft. The Pearson correlation coefficient, $r$, was calculated for each station. Bold correlation coefficient values indicate significant correlation at the 95% confidence level. Bottom line values show the averages of RMSD, mean difference and correlation coefficient over all stations.

| Period | Station | CS2 | | | CS2orbits | | |
| --- | --- | --- | --- | --- | --- | --- | --- |
| | | RMSD | Mean difference | r | RMSD | Mean difference | r |
| | | [m] | [m] | | [m] | [m] | |
| 2010-2011 | Outer Shelf | 0.83 | -0.68 | 0.61 | 0.94 | -0.65 | **0.39** |
| 2011-2012 | Outer Shelf | 0.58 | -0.02 | 0.29 | 0.71 | -0.06 | **0.38** |
| 2013-2014 | 1893 | 0.23 | -0.06 | 0.71 | 0.22 | -0.02 | **0.82** |
| | Taymyr | 0.68 | -0.53 | 0.53 | 0.71 | -0.47 | **0.43** |
| | Kotelnyy | 0.61 | -0.41 | 0.74 | 0.61 | -0.46 | **0.68** |
| | Vilkitzkii | 0.24 | -0.02 | 0.46 | 0.44 | -0.35 | **0.73** |
| 2014-2015 | 1893 | 0.55 | -0.46 | 0.46 | 0.51 | -0.39 | **0.55** |
| | Taymyr | 0.32 | -0.27 | 0.70 | 0.41 | -0.28 | **0.54** |
| 2014-2016 | Vilkitzkii1 | 0.40 | -0.02 | 0.10 | 0.57 | 0.02 | -0.06 |
| | Vilkitzkii3 | 0.40 | -0.19 | 0.37 | 0.58 | -0.14 | 0.21 |
| **2010-2016** | **Mean** | 0.48 | -0.27 | 0.50 | 0.57 | -0.28 | 0.47 |





**Table 3.** Statistics of the comparison of gridded weekly mean CS2SMOS draft data with VAL mean sea ice draft for the period from 2010 to 2016. RMSD and mean difference were calculated for the differences of CS2SMOS minus VAL mean sea ice draft. The Pearson correlation coefficient, *r*, was calculated for each station. Bold correlation coefficient values indicate significant correlation at the 95% confidence level. Bottom line values show the averages of RMSD, mean difference and correlation coefficient over all stations.

| Period | Station | CS2SMOS | | |
| --- | --- | --- | --- | --- |
| | | RMSD [m] | Mean difference [m] | r |
| 2010-2011 | Outer Shelf | 0.88 | -0.70 | 0.41 |
| 2011-2012 | Outer Shelf | 0.48 | -0.07 | **0.72** |
| 2013-2014 | 1893 | 0.32 | -0.17 | **0.70** |
| | Taymyr | 0.92 | -0.76 | **0.51** |
| | Kotelnyy | 0.73 | -0.64 | **0.92** |
| | Vilkitzkii | 0.29 | -0.18 | **0.78** |
| 2014-2015 | 1893 | 0.46 | -0.42 | **0.80** |
| | Taymyr | 0.40 | -0.36 | **0.77** |
| 2014-2016 | Vilkitzkii1 | 0.50 | -0.24 | 0.10 |
| | Vilkitzkii3 | 0.59 | -0.41 | **0.42** |
| **2010-2016** | **Mean** | 0.56 | -0.39 | 0.61 |



**Table 4.** Statistics for the comparison between gridded CS2, CS2orbit and gridded CS2SMOS mean sea ice draft with modal VAL sea ice draft from the period from 2013 to 2015. Due to the low temporal resolution of the ADCP-derived VAL data, modal sea ice draft was only calculated for ULS data. RMSD and mean difference were calculated for the difference between mean satellite minus modal VAL data. The Pearson correlation coefficient, $r$, was calculated for each of the four VAL data sets. The values show the mean of RMSD, mean difference and $r$ over the four VAL data sets. Bold mean correlation coefficients indicate significance of all four correlation coefficients at the 95% confidence level. None of the correlations was significant for the CS2 data.

|                       | CS2  | CS2orbit | CS2SMOS |
|-----------------------|------|----------|---------|
| RMSD [m]              | 0.25 | 0.30     | 0.21    |
| Mean difference [m]   | 0.05 | 0.06     | -0.05   |
| $r$                   | 0.61 | **0.63** | **0.77** |