# Peer review of "Satellite-based sea ice thickness changes in the Laptev Sea from 2002 to 2017: Comparison to mooring observations"

_The Cryosphere, 2019_

## Referee Comment (RC1) · Anonymous Referee #1 · 11 Feb 2020

**Satellite-based sea ice thickness changes in the Laptev Sea from 2002 to 2017: Comparison to mooring observations**

Hans Jakob Belter, Thomas Krumpen, Stefan Hendricks, Jens Hoelemann, Markus Janout, Robert Ricker, and Christian Haas

https://doi.org/10.5194/tc-2019-307

**11 Feb 2020**

**General comments**

ESA CCI-2 project has produced SIT CDR over the Arctic based on ENVISAT and CryoSat-2 (CS2) radar altimeter data. The SIT CDR covers the period from 2002 to 2017 and winter months from Oct to Apr, and includes both monthly gridded data (25 km) and SIT data along orbit tracks. The authors of this paper have compared this SIT CDR to in-situ sea ice draft measurements (VAL data) by ULS and ADCP sensors in the Laptev Sea. The SIT CDR data was converted for the draft data for this comparison. In addition, also SIT data from merged SMOS-CS2 product converted to draft was compared to the VAL data, but this study has a minor part in the paper (which is ok). The draft data from ADCPs (accuracy ±0.1 m) cover years 2003-2016 and more accurate (accuracy ±0.05 m) ULS draft data is for 2013-2015. The main result of the study is very interesting: in the Laptev Sea, where first-year ice mainly occurs, the SIT CDR data agrees better with the modal sea ice draft measured by ULS and ADCP, and it overestimates drafts below 0.7 m and underestimates drafts over 1.3 m with this underestimation increasing with increasing draft. The modal draft (and SIT) is related to sea ice thermodynamic growth, i.e., SIT/draft of level ice. This seriously hampers (if I understood correctly) investigating temporal SIT trends in the Laptev Sea based on the CCI-2 SIT CDR (or any other SIT record based on the radar altimeter data). A comparison between the validation data (VAL) and SIT CDR (SAT-VAL difference) showed linear trends in the SAT-VAL difference in different VAL draft ranges (0-1 m, 1-2 m, 2-3 m), but these trends could be robustly attributed to intermission bias between ENVISAT and CS2. Therefore, the accuracy of the CCI-2 SIT CDR was found to be stable over time, but its uncertainty is different from ENVISAT and CS2 periods. The SIT CDR were used to briefly investigate SIT anomaly trends (Figure 2). Trends were found, but the only trend during the CS2 period is statistically significant. It was emphasized that these trends need to be investigated further due to short periods and high uncertainties of the SIT products.

In general, the study set up with data acquisitions and data processing is sound, as are methods for the data analyses. The conclusions in the paper are well justified by the results from the data analyses. This paper should be of interest for the scientists producing or analyzing SIT records from RA data, and it fits nicely to the scope of TC. However, I have following general comments which could improve the paper:

The authors should discuss whether their main results are also applied to other Arctic areas dominated by FYI, like Kara Sea, and point out clearly whether the current CCI-2 CDR can be used to investigate SIT trends over FYI dominated oceans; in Figure 2 only SIT anomaly trend from the CS2 period was statistically significant.

In Introduction Section you could review what is current understanding on the accuracy/quality of the CCI-2 SIT CDR: it seems this has been investigated at least by Kern et al. 2018. Are there any other studies, especially in peer-reviewed journals? You could also review similar other studies: comparisons between RA SIT records and sonar draft data. What is the typical relationship(s) between sonar and RA drafts over MYI?

A short section describing typical sea ice conditions and typical progress of sea ice season in the Laptev Sea would be good addition to the paper. How much there can be MYI in the Laptev Sea?

Can there be large areas of grounded landfast ice for which the used freeboard to SIT conversion is not valid, and thus, could have an effect on your results?

Sections 2-4 should have short introductions about their content and focus.

The processing of ADCP data to the sea ice draft is based on reference (Belter et al., 2019b), but this is paper under review; so it is possible that it may not ever get published, or at least at time of possible publication of this paper this reference is not available. Is it possible to include this ADCP processing method (summary) as Appendix here? Are there any conference papers, web-pages, etc., you could also have as references?

In Section 2 you could have a sub-Section which describes how different datasets are processed to match each other. Now this information is scattered in sub-Sections describing the datasets. Also include a Table which summarizes datasets: spatial and temporal resolutions, accuracies, etc.

How many pixels there are in the gridded datasets over the Laptev Sea? This is relevant to Figure 2. The gridded (25 km) SIT data were selected from and an area of 25 km radius around a sonar mooring, thus at maximum four pixels were selected? You should give these kind details on the dataset matching Section.

The uncertainty of the sea ice draft calculated from the CCI-2 CDR SIT data is estimated with (1)

$d_{unc}=d/SIT*SIT_{unc}$

But $d = SIT$ – freeboard, so $d_{unc}$ could be sqrt($SIT_{unc}$^2+$fb_{unc}$^2)? Well SIT and fb are correlated…How about if you estimate $d_{unc}$ with typical uncertainties of all parameters in the equation $d = SIT – fb$; e.g. snow thickness, would you end up a same figure as with (1)?

Section 3.3.2 Merged CS2SMOS sea ice draft contains also a summary of all results; this should be in its own sub-Section.

Section 4.4, Taymyr 2013/2014 case, is under 'Discussion', but it includes data processing and analyses, these could be under 'Results', also the data processing methods would fit better to Section 2. Why this very important case study which reveals that the CDR SIT correspond modal sea ice draft, and not the mean draft, was not repeated with any ADCP dataset? This would be very important so that we would see consistency of this conclusion. This case study could be also described with more details in Introduction, now only one sentence.

Tables 1-3 show averages of statistical parameters from different mooring locations. I am not sure this is meaningful, what an average correlation coefficient really tells us here? I think better would be here to combine all datasets together and then calculate RMSD, mean difference and r.

**Specific comments**

**1 Introduction**

line 21: "While knowledge about SIC is widely available it provides limited insight into overall sea ice changes."

You could include reference(s) to SIC records, like OSI SAF ones.

l. 26: "Satellite remote sensing of SIC started in the 1970s with passive microwave sensors (Parkinson et al., 1999) and was further developed, updated and improved by multiple follow-on missions (Comiso and Nishio, 2008; Cavalieri and l. Parkinson, 2012) until today."

Some newer references would be nice, like:

Lavergne et al., Version 2 of the EUMETSAT OSI SAF and ESA CCI sea-ice concentration climate data records, The Cryosphere, 13, 49–78, 2019

l. 36: "the impact of snow radar backscatter"

the impact of snow on radar backscatter?

Explain that both gridded and orbit track SIT CDRs are used in your study.

l. 64: 'Taymyr mooring'

at this point a reader does not know what this Taymyr mooring is

**2 Data and methods**

Can you explain why ADCPs where not moored at some locations in different years?

Figure 1: you could mask land out; add color scale for the water depth; show boundaries of the Laptev Sea.

It would be interesting to see what is the typical variation of the sonar draft during a day, week and month. A figure about a time series of sea ice draft from some ADCP location would be nice.

How sonar draft data was processed to a monthly scale, just averaging all datapoints?

Are there any peer-reviewed journal papers that could be put as reference to CCI-2 SIT CDR in Section 2.2.1? A figure about monthly gridded SIT over the Laptev Sea would be interesting see; also what it typical SIT spatial variation over the Laptev Sea in this monthly product? How many pixels there are over the Laptev Sea?

l. 118: "Although Paul et al. (2018) minimized the inter-mission sea ice freeboard biases for the basin average, ENVISAT freeboards in multi-year ice (MYI) regions are still thinner than CS2 freeboards, while ENVISAT provides thicker freeboards than CS2 in regions that are dominated by FYI."

Give some figures; how much thinner and thicker.

l. 127: "a weighted mean sea ice draft value."

What is this weighting?

Section 2.2.2

Give typical uncertainty in a single SIT orbit value. How many SIT points are typically averaged within 25 km radius and in a daily scale? What is the uncertainty of this average?

**3. Results**

Comment correlations shown in Figure 2 in Section 3.1.

Did you investigate SIT anomalies in different part of the Laptev Sea? I think AARI uses Eastern and Western Laptev Sea regions. Is it possible to compare the SIT anomaly trends to any other study/data source? E.g. based on AARI ice charts? Are the trends related to polynya activity (extent, ice production) in the Laptev Sea?

Figure 3(b) is not commented/discussed in the text; e.g. symmetry/normality of the pdf?

Section 3.2 title could be "Gridded monthly sea ice draft".

Section 3.3 title "Higher temporal resolution satellite products" is not good, what is this 'higher'?

"Daily and weekly sea ice draft products"?

In Figure 4 it is difficult to see grey crosses. Maybe these single datapoints can be removed and instead describe in the text how many RA draft datapoints were typically in each VAL draft interval.

l. 214: "It also confirms the intermission biases between ENVISAT and CS2 that were published by Paul et al. (2018)."

Please give out these biases here.

l. 227: "Consequently the underestimation of sea ice draft with increasing thickness is largest for CS2SMOS because of the larger uncertainties of SMOS over thicker sea ice."

But for thicker ice CS2MOS SIT comes only from CS2 data? If so then SMOS uncertainty should have no effect here.

**4 Discussion**

Figure 6(b) is not discussed in the text.

What data is ICETrack using? SAR imagery? Describe in the text.

l. 346: "That means that, for investigations into the sea ice cover in the Laptev Sea it is important to be aware that sea ice can persist some time after the presented satellites stop providing SIT data."

How about before the winter in late summer? Phrasing here: satellites do not provide SIT data, but your data processing methods do, i.e. SIT estimation is not possible (at least currently) for summer/melting season.

**5 Conclusions**

l. 350: "The ESAs CCI-2 gridded SIT CDR covers a period from 2002 to 2017 and has been validated for multiple regions around the Polar regions of the Earth."

A short summary about the results of these validation activities would be good here. Later in Conclusions you could summarize what new insight in the accuracy of the CCI-2 SIT CDR your study resulted.

l. 378: "Therefore, improvements in the processing of radar altimetry data are required for the estimation of surface roughness but also for the parametrizations of snow depth and densities of snow and ice."

How surface roughness would be utilized in the freeboard tracking? How about different freeboard trackers for different ice types, like FYI and MYI?

l. 383: "Furthermore would continuous long-term SIT measurements in the Laptev Sea provide much needed information on deformation processes."

Is this supposed to be an interrogative clause? Or "Furthermore, continuous long-term SIT measurements in the Laptev Sea would provide…"?

---

## Referee Comment (RC2) · Anonymous Referee #2 · 13 Feb 2020

In the manuscript the authors inter compare satellite-based sea ice thickness retried from ENVISAT and Cryosat2 data with the sea ice thickness retrieved from ADCP and ULS measurements in the Laptev Sea. The comparison give a new insights into the satellite-based sea ice thickness data and is of interest for sea ice community.

General comments:

1. The 'Data and method' section lacks important details. The data section only briefly introduces different data sets. It is not clear how many measurements were compared and how the mean Laptev Sea sea ice thickness from gridded data was calculated. Section 4.1. and 4.2 describing data limitations can be moved to the 'Data and meth-

ods' to provide the reader with valuable information before introducing the results.

2. The discussion can be elaborated. - The Anabar, Khatana and Lena stations a located in the area of polynya formation. Are polynya events taken into account in SAT and VAL data? Do the polynya events affect your comparison between SAT and VAL monthly mean? - One of the main finding shows that SAT data represents modal sea ice thickness rather than mean. What is a possible explanation? - The SAT-VAL difference depend on sea ice thickness. It there a seasonal change in this difference? I suggest that a scatter plot with seasonal cycle might be informative. - Section 4.4 introduces new data, method and results. Would it make more sense to restructure it and add a subsection to methods and results? Why other data from ADCP and ULS is not shown? Does it confirm you findings?

Specific comments:

Line 160: 'The ESA CCI-2 SIT CDR shows an overall thinning of sea ice in the Laptev Sea between 2002 and 2017.' The sentence about is too strong. The error of the overall trend is as large as trend. Also the significance of the trend is quite low. The black line rather shows that there is no changes in sea ice thickness.

Lines 161-162: How is the Laptev Sea defined? Please show the region used for SIT anomaly calculation in Figure 1.

Line 210: 'ENVISATorbit data shows a higher average RMSD, stronger average underestimation of VAL sea ice draft and much lower average correlation with VAL sea ice drafts compared to the gridded ENVISAT data' Is there an explanation? Why does the orbit data which supposed to be closer to the VAL measurement shows worse statistical characteristics?

Lines 282-283: 'The seasonal biases between ENVISAT and CS2 need to be considered for the temporal development of the Laptev Sea SAT-VAL differences between the two periods'. Please elaborate. Are those biases considered in this study?

Lines 343-348: It is worth mentioning that ULS provide sea ice draft measurements after the onset of melt. However it is not a real finding that there is sea ice in the Laptev Sea in June-July. Please consider reformulating.

Line 359: 'The presented satellite products represent similar sea ice drafts differently.' I am not sure the meaning is clear. Do you mean identical sea ice draft or sea ice draft of similar thickness, e.g. within presented bins?

Technical comments:

Page 1 line 24: sea ice system –> sea ice state?

Page 2 Line 43: a space after '(ULS)' is missing

Line 132: a space after 'ENVISAT' is missing

Figure 2: It seems that colors of he legend in the upper left corner are mixed up. The negative trend should be the ENVISAT one.

Figure 7: The scale on the sea ice draft axis is missing

---

## Referee Comment (RC3) · Anonymous Referee #3 · 13 Feb 2020

In this paper the authors validate the sea ice thickness (SIT) climate data record (CDR) produced from the Envisat and Cryosat-2 radar altimeter data in the frame of the ESA Sea Ice Climate Change Initiative Phase 2 project. The authors compare the sea ice draft estimated from the sea ice freeboard and thickness retrievals with the draft measurements of the sonar-based ULS and ADCP in the Laptev Sea over the period 2003 to 2016. It is found that the satellite-derived draft is overestimated (underestimated) over thin (thick) ice, but the performance of the monthly gridded CCI-2 SIT CDR is stable over time. Analysis of the daily mean sea ice draft time series derived from along-track Cryosat-2 data shown that it is in agreement rather with the modal sea ice draft derived from ULS measurements that is attributed to inability of radar altimetry to

reproduce changes in the SIT caused by sea ice dynamics.

The paper provides extension of the SIT products' validation activity to the seas of the Russian Arctic, and is of interest in terms of interpretation and generation of the SIT datasets derived from satellite radar altimetry. The paper worth publication, but several following comments should be considered for improving the manuscript.

Specific comments: Line 12: This phrase does not fully correspond to the results presented in the paper. Overestimation (underestimation) of sea ice draft for thin ice below 0.7 m (thick ice above 1.3 m) is indicated from comparison of the mean values, but not with respect to the modal draft. Line 40: Authors could also note that in (Kern et al., 2018) the airborne Operational Ice Bridge data were used for validation of the satellite product as well. Section 2.1: I guess that open water was excluded from the sonar-based measurements. If so, please, mention it in the text. Line 101: The phrase 'bottom track mode measurements of surface and error velocity' sounds not clear. Although paper by Belter et al. (2019b) will, I guess, describe details of the methodology, some clarifications on what is, e.g., 'error velocity' would be helpful. Line 125: This way of estimating draft uncertainty is applicable if SIT uncertainty accounts for the sources of freeboard uncertainty. If so, please, mention it in the text. From the other side, since the authors do not use this draft uncertainty further in the analysis, it is not clear what it was estimated for. Line 127: It is not clear how authors calculate weighted mean values. Possibly this weighting account for the distance between grid center and mooring location? If so, please clarify it. Line 133: As a frequency of the orbit tracks that pass over the mooring sites the authors specify 'four overflights'. However Envisat and Cryosat-2 have different orbit inclinations and this frequency should be different for these missions. Section 3: I suggest the authors to change structure of this section: to combine sections 3.2 and 3.3 in one section 3.2 with the title, for example, 'Validation of CCI-2 products', and with the subsections '3.2.1 Gridded CCI-2 sea ice draft'(currently section 3.2), '3.2.2 Orbit CCI-2 sea ice draft' (currently section 3.3.1), and '3.3.3 Intercomparison of CCI-2 and merged CS2SMOS drafts' (currently section

3.3.2). Then the accordingly revised text from the first paragraph of the current section 3.3 could be moved to the beginning of new section 3.2 Line 228: I guess that this enhanced underestimation of thick ice by CS2SMOS data is observed because for some bins corresponding to thick ice the CS2SMOS product is the only available data (as I can see from Figure 5). It means that for these bins CS2SMOS product is generated only from the SMOS measurements. If so, this could be explained in the text. Line 285: The reference (Paul et al., 2018) is not appropriate here. Paul et al., 2018 do not provide regional estimates of the differences between SIT derived from ENVISAT and CS2 data. Line 315: The indicated trends are small, that supports the conclusion that the gridded CCI-2 CDR is stable over considered period. However Fig.6 shows that these trends might be caused not only by the intermission differences. The trends for thickness ranges 0 to 1 m and 1 to 2 m looks negative even separately for Envisat and CS2 data as well as for combined dataset. For thickness range 2 to 3 m two overlapping points in 2011 shows that Envisat rather overestimate sea ice draft as compared to CS2 as well as for thinner ice. Line 380: It can be noted here that not only snow depth, but specifically snow properties that influence the location of the main scattering horizon are a major source for uncertainty in the freeboard retrieval process.

Technical comments: Line 79: I suggest to replace 'approaches' by 'instruments', otherwise one may interpret it that both ADCP and ULS data are processed by two methods. Line 101: Abbreviation 'BT' is not needed here as it is not used further in the text. Figure 2: Colours of the first and second lines indicating trend values should be switched Line 231: I suggest to reformulate this sentence: 'While individual stations deviate from this average the overall tendency indicates a dependency of the agreement between monthly mean gridded CCI-2 and VAL sea ice draft on sea ice thickness.' Line 332: In the 'newly formed FYI ice' the word 'ice' is not needed. Line 383: The sentence 'Furthermore . . .' sounds not clear. Please, consider revising. Table 4: In the captions to the Table it is noted that the statistical parameters 'were calculated for the four VAL data sets '. However this table presents the results only for two stations with ULS measurements: Taymyr and 1893.

---

## Author Comment (AC1) · 16 Mar 2020

Point-by-point response to editor and reviewer concerns by
corresponding author: H. Jakob Belter
March 16, 2020

**tc-2019-307:**

**Satellite-based sea ice thickness changes in the Laptev Sea from 2002 to 2017: Comparison to mooring observations**

https://doi.org/10.5194/tc-2019-307:

Belter, H. Jakob, Thomas Krumpen, Stefan Hendricks, Jens A. Hoelemann, Markus A. Janout, Robert Ricker, Christian Haas

Dear anonymous Reviewer #1,

on behalf of all authors, I would like to thank you for your detailed and constructive comments. In the following you can find a point-by-point response to your comments. We really have the feeling that your insights helped improve the manuscript and we hope that all your concerns have been answered to your satisfaction. We would also like to refer you to the responses to the other reviewers for more improvements and changes to the manuscript

General comments:

- *(1) The authors should discuss whether their main results are also applied to other Arctic areas dominated by FYI, like Kara Sea, and point out clearly whether the current CCI-2 CDR can be used to investigate SIT trends over FYI dominated oceans, in Figure 2 only SIT anomaly trend from the CS2 period was statistically significant.*

  -Response:

  This is a really good question that unfortunately can not be answered given the data limitations we are facing in the Russian Shelf Seas. We do not see why the presented results should not be confirmed for similar FYI-dominated shelf seas, however, since we only happen to have moorings in the Laptev Sea we can not provide any proof. Therefore, this study is focused on the region where data is available to us - the Laptev Sea. Our conclusion clearly states that these results only concern the Laptev Sea, however, as we are also providing suggestions and 'to-dos' for further algorithm development we reckon that agreement between satellite and VAL data will be improved across the Arctic. However, without actual in situ observations to validate the satellite products in other regions of the Arctic this remains speculation. In fact, having this newly acquired sonar-based validation data set for the Laptev Sea is already a big step towards analysing regional differences in the performance of the available satellite products. We are certainly hoping for more data sets from other regions but also long-term measurements of similar quality as the unique ULS sea ice draft time series in Fram Strait (NPI and Hansen et al.,2013) for future validation purposes. As for your second comment, longer time series are also the aim for the CCI-2 SIT CDRs, as they will help strengthen the statistical significance of possible satellite-based SIT anomaly trends (Fig. 2).

  -Changes:

  No changes.

- *(2) In Introduction Section you could review what is current understanding on the accuracy/quality of the CCI-2 SIT CDR: it seems this has been investigated at least by Kern et al. 2018. Are there any other studies, especially in peer-reviewed journals? You could also review similar other studies: comparisons between RA SIT records and sonar draft data. What is the typical relationship(s) between sonar and RA drafts over MYI?*

  Response:

  In order to avoid a lengthy introduction we decide to combine this request for an additional review of already existing validation results with your later comment in the 'Conclusion'. Rather than introducing validation results over MYI here we move them to the 'Conclusion'.

Changes:
No changes in the 'Introduction' but a short additional review of previous results and a comparison between MY and FYI results in the 'Conclusion'.

- *(3) A short section describing typical sea ice conditions and typical progress of sea ice season in the Laptev Sea would be good addition to the paper. How much there can be MYI in the Laptev Sea? Can there be large areas of grounded landfast ice for which the used freeboard to SIT conversion is not valid, and thus, could have an effect on your results?*
  -Response:
  Sea ice in the Laptev Sea is mostly FYI, however, fast ice is also present in the coastal regions. As for the influence of fast ice on the satellite SIT thickness: We consider the lack of leads and thus the lack of constraints for the instantaneous sea-surface height the primary issue for SIT retrieval in coastal fast ice regions. Therefore any valid freeboard point in the CCI SIT dataset is discarded if it is further away than 100 km from the next sea-surface height tie points. For the freeboard to thickness conversion we assume that the main sea ice mass is in isostatic compensation with the exception of the grounded anchor points. However, for the satellite SIT data we are comparing to the mooring data this shouldn't be a concern since the mooring locations are far enough away from the fast ice edge. Lena, Anabar and Khatanga stations are the farthest south but rather influenced by the polynyas than the fast ice.
  -Changes:
  We added a short paragraph to introduce the general conditions of the Laptev Sea sea ice cover to the 'Introduction' (**LINES 52-57**).

- *(4) Sections 2-4 should have short introductions about their content and focus.*
  -Response:
  In order to keep this paper as short as possible we tried to clearly distinguish between the different sections. We also made sure to specify our objectives in the 'Introduction' and used multiple subsections in the individual sections so that it is clear what we are presenting and when. We think that with your suggestions for new subsection titles this is even clearer now. We also know that separate introductions into the sections are a personal preference and we hope that you can condone that we prefer to leave these additional introductions out to not interrupt the flow of the text.
  -Changes:
  Title changes in the subsections of the 'Results' section (**LINES 215, 216, 253, 260**).

- *(5) The processing of ADCP data to the sea ice draft is based on reference*

*(Belter et al., 2019b), but this is paper under review; so it is possible that it may not ever get published, or at least at time of possible publication of this paper this reference is not available. Is it possible to include this ADCP processing method (summary) as Appendix here? Are there any conference papers, web-pages, etc., you could also have as references?*
-Response:

You are quite right, unfortunately, the corresponding method paper is still under review. We hope that the below mentioned changes and additions are sufficient for now and keep our fingers crossed that the method paper will be accepted soon.

-Changes:

For now we published the ADCP-derived sea ice draft time series for all the stations that have been used in this manuscript. We also added a short summary of the processing steps for the derivation of sea ice draft from upward-looking ADCPs in the Laptev Sea to these data sets (Belter et al.,2020a, **LINE 116**) and refer to it in the 'Data and methods' section of this manuscript. In response to Reviewer #3 we extended the subsection on this new method in the manuscript as well **LINES 112-116**.

- *(6) In Section 2 you could have a sub-Section which describes how different datasets are processed to match each other. Now this information is scattered in sub-Sections describing the datasets. Also include a Table which summarizes datasets: spatial and temporal resolutions, accuracies, etc.*
  -Response:

  Thank you for this helpful comment. We summarized how the different data sets are processed to be comparable to the VAL data in a subsection below the introduction of all the satellite data sets ('Data and methods' section). As for the suggested table, we do not think that a table is the right way to go here, although we could certainly summarize the temporal and spatial resolutions, accuracies and uncertainties are a little less clear because they vary on temporal but also spatial scales. Every CCI-2 grid point has its own uncertainty value. Orbit data for example is only available when the trajectory of the satellite coincides with the 25 km area around the mooring, this is something that can happen six or seven times in one month and only 3 times in others. Although we agree that a table supports comparability of the individual products some of them would not fit in or would have unclear values for the selected parameters (spatial, temporal resolution, accuracies, etc.). We would therefore leave the description of the data sets the way they are, clearly mentioned in the individual paragraphs of the different products. We hope you agree.

  -Changes:

Summary paragraph for satellite data processing following the introduction of the satellite data sets **LINES 161-171**).

- *(7) How many pixels there are in the gridded datasets over the Laptev Sea? This is relevant to Figure 2. The gridded (25 km) SIT data were selected from and an area of 25 km radius around a sonar mooring, thus at maximum four pixels were selected? You should give these kind details on the dataset matching Section.*
-Response:
We are not quite sure whether adding this information is really necessary. As you rightly mentioned in your comment when selecting gridded data (25 km grid) from within a radius of 25 km around the mooring the maximum number of values is four. This is the case for the gridded CCI-2 data sets ENVISAT and CS2 but also the CS2SMOS one. For the orbit data this number is significantly higher. We therefore would have to mention this detail about the number of data points for each of the presented satellite data sets separately, which is contradicting your previous comment ((6)) about combining the information about the processing into on paragraph rather than scattering it over the individual data set paragraphs. We agree with your previous comment that one summary is the better choice and with the clear information of what the data sets look like in terms of resolution the reader can see how many data points combine for the satellite-derived mean draft value. The same holds for the Fig. 2 data, we added the respective area from which the data was selected to Fig. 1 which helps understand the number of data points that go into the results presented in Fig. 2 the grid resolution was also added to the caption of Fig. 2. We hope you agree with this assessment and the changes that we made.
-Changes:
Addition to Fig. 2 caption.

- *(8) The uncertainty of the sea ice draft calculated from the CCI-2 CDR SIT data is estimated with (1)*
  *dunc = d/SIT · SITunc*
  *But d = SIT - freeboard, so dunc could be sqrt(SITunc$^2$ + fbunc$^2$)? Well SIT and fb are correlated. How about if you estimate dunc with typical uncertainties of all parameters in the equation d = SIT − fb e.g. snow thickness, would you end up a same figure as with (1)?*
-Response:
You are right that the CCI-2 draft uncertainty should be calculated differently, however, Reviewer#3 rightly mentioned that we are not really using draft uncertainty anywhere in the paper except Fig. 7. We therefore removed that entire part about draft uncertainty and the uncertainty bars that were

visible in Fig. 7 before. Since uncertainty is an important part of an analysis such as this one we provide typical SIT uncertainty information for each of the discussed satellite products in the respective 'Data and methods' satellite data subsections.
-Changes:
Uncertainty equation and uncertainty bars in Fig. 7 have been removed. Information about uncertainties have been added to the respective 'Data and methods' satellite data subsections.

- *(9) Section 3.3.2 Merged CS2SMOS sea ice draft contains also a summary of all results; this should be in its own sub-Section.*
  -Response:
  We added a free line after the Section 3.3.2. to show that the summary below is the summary for the entire 'Results' chapter rather than the 3.3.2. Section.
  -Changes:
  Addition of extra line after **LINE 272**.

- *(10) Section 4.4, Taymyr 2013/2014 case, is under 'Discussion', but it includes data processing and analyses, these could be under 'Results', also the data processing methods would fit better to Section 2. Why this very important case study which reveals that the CDR SIT correspond modal sea ice draft, and not the mean draft, was not repeated with any ADCP dataset? This would be very important so that we would see consistency of this conclusion. This case study could be also described with more details in Introduction, now only one sentence.*
  -Response:
  Thank you for commenting on this case study, we discussed the question whether this should be part of the 'Results' or 'Discussion' section among the authors as well and found that this case study is not so much a new result at this point but an example that highlights and explains the results we already presented. Rather than leaving the reader with the comment: 'overestimation by the satellites for thicknesses below 0.7 m and underestimation for thicknesses above 1.5 m', we discuss how these results can be explained and what satellite draft values really show. This case study and more specifically the comparison between modal VAL and mean satellite drafts are the means to further explain and interpret the results shown in Section 3. The analysis of ICETrack data is done to determine possible reasons why satellite and VAL draft do not agree well, especially when VAL data suggests large daily mean draft values, like between Jan and Mar 2014 at the Taymyr mooring. We therefore would like to keep the case study subsection at the end of the 'Discussion' as it is right now. As for the repetition of this case study with ADCP data: Due to the low temporal resolution of the ADCP measurements

(mostly hourly and half-hourly values) the number of values per day is not large enough to compute the ADCP ice draft modes reliably. We therefore focused on the four draft time series that are based on high resolution ULS data (2013-2015) and provide reliable modal sea ice draft values. The Taymyr 2013/2014 results showing the better agreement with modal VAL data are confirmed by the other three time series that were analysed.

-Changes:

We added a respective sentence on why the comparison to modal values was only done with ULS data to the text (**LINE 364-366**), however, it is also mentioned in the caption of Tab. 4.

- *(11) Tables 1-3 show averages of statistical parameters from different mooring locations. I am not sure this is meaningful, what an average correlation coefficient really tells us here? I think better would be here to combine all datasets together and then calculate RMSD, mean difference and r.*

  -Response:

  We agree with you, that averages of the correlation coefficients do not tell us anything about how good the agreement is between individual VAL and satellite data, however, we are providing the values of RMSD, mean difference and correlation coefficient for each of the stations individually and add those averages over all stations only for additional information and comparison between the different satellite data sets. With the correlation coefficients available for the individual stations as well we feel that it is clearly displayed what these averages are and it is legitimate to show them here. In case of the mean difference and RMSD these average values actually tells us how different VAL data is from the respective satellite data, on average. While average correlation coefficient might only provide a measure to compare the different satellite eras, combining all data sets to calculate an overall correlation coefficient might not be meaningful either. There are a couple of time series that were recorded simultaneously which means that in order to provide an all data correlation coefficient we either need to leave out time series when more than one is available in some years or we need to combine data that was recorded at the same time which involves more averaging and altering of the data. However, calculating all-data-versions of the statistical parameters would certainly be the right way in cases where the entire time series was recorded at the same position.

  -Changes:

  We hope you agree that we clearly state that the averages of the correlation coefficients are nothing more than averages over all stations and do not provide information about how good the general agreement between VAL and the respective satellite data is. No additional changes have been made here.

Specific comments:

- *(1) line 21: 'While knowledge about SIC is widely available it provides limited insight into overall sea ice changes.'*
  *You could include reference(s) to SIC records, like OSI SAF ones.*
  *26: 'Satellite remote sensing of SIC started in the 1970s with passive microwave sensors (Parkinson et al., 1999) and was further developed, updated and improved by multiple follow-on missions (Comiso and Nishio, 2008; Cavalieri and l. Parkinson, 2012) until today.'*
  *Some newer references would be nice, like:*
  *Lavergne et al., Version 2 of the EUMETSAT OSI SAF and ESA CCI sea-ice concentration climate data records, The Cryosphere, 13, 49-78, 2019*
  -Response:
  Thank you for this suggestion. We added the recommended reference to provide some more recent publications.
  -Changes:
  Citation added (**LINE 28**).

- *(2) 36: 'the impact of snow radar backscatter'*
  *the impact of snow on radar backscatter?*
  -Response:
  Changed.
  -Changes:
  **LINE 37**

- *(3) Explain that both gridded and orbit track SIT CDRs are used in your study.*
  -Response:
  We added a sentence to the objectives paragraph of the 'Introduction' to indicate that we will also compare VAL data to higher temporal resolution satellite products. We do not mention these higher resolution products here since they are properly explained in the 'Data and methods' section right below.
  -Changes:
  **LINES 76-77**

- *(4) 64: 'Taymyr mooring', at this point a reader does not know what this Taymyr mooring is*
  -Response:
  We added a link to Fig. 1 (the map) so that the reader can find the mooring location here.
  -Changes:
  Reference to Fig. 1 (**LINE 78**).

- *(5) Can you explain why ADCPs where not moored at some locations in different years?*
  -Response:
  There are a number of reasons why some locations weren't visited more often. For one, most of these mooring locations are within the Russian EEZ, which requires a permission for the deployment and recovery of moorings from the Russian government. Secondly, were the expeditions based on multiple different research proposals and therefore varying research questions. While one location might have been interesting for one project it was not for one of the following projects. In the end, none of the available ADCPs were specifically deployed for the purpose of measuring sea ice draft therefore it was not a priority to generate a long-term time series at a specific location.
  -Changes:
  No changes required.

- *(6) Figure 1: you could mask land out; add color scale for the water depth; show boundaries of the Laptev Sea.*
  -Response:
  We kept the land, but added a proper color scale for bathymetry of land and ocean.
  -Changes:
  Colour scale added to Fig. 1.

- *(7) It would be interesting to see what is the typical variation of the sonar draft during a day, week and month. A figure about a time series of sea ice draft from some ADCP location would be nice.*
  -Response:
  We agree that the variation of sea ice draft is very interesting especially since it is not visible from either the daily mean VAL values nor the satellite data, however, since this study is focused on validating the ESA CCI-2 SIT CDR and other satellite SIT products we feel that this extra figure would be outside the scope of this study and simply to much information. An example of the variation on a monthly scale is given in Fig. 7 (the Taymyr case study) and we are happy to provide the high frequency (1 Hz) sea ice draft time series (also for the Taymyr case) below (Fig. 1).
  -Changes:
  No changes to the manuscript.

- *(8) How sonar draft data was processed to a monthly scale, just averaging all datapoints?*
  -Response:
  You are right, sonar draft was simply averaged over the respective month,

[Figure]

Figure 1: ULS sea ice draft at the Taymyr mooring (2013-2014). Grey line shows raw (1 Hz sampling frequency), orange line shows daily mean sea ice draft data.

however, we only saved a monthly mean value when data was available for at least 50% ('number of data points'-wise) of the month.
-Changes:
An additional paragraph was added to the 'Data and methods' section (**LINES 121-124**, also in response to Reviewer #3)

- *(9) Are there any peer-reviewed journal papers that could be put as reference to CCI-2 SIT CDR in Section 2.2.1? A figure about monthly gridded SIT over the Laptev Sea would be interesting see also what it typical SIT spatial variation over the Laptev Sea in this monthly product? How many pixels there are over the Laptev Sea?*
  -Response:
  Unfortunately, there are no peer-reviewed publications for the CCI-2 SIT CDRs. Only publications about radar altimetry freeboard are available and have been cited here (Paul et al., 2018). The citation for the data sets themselves are in the Section 2.2.1 text as well.
  Arctic-wide sea ice thickness data is available (at least for the CS2 period) on the seaiceportal (https://data.meereisportal.de/gallery/).
  -Changes:
  No changes to the text required.

- *(10) 118: 'Although Paul et al. (2018) minimized the inter-mission sea ice freeboard biases for the basin average, ENVISAT freeboards in multi-year ice*

*(MYI) regions are still thinner than CS2 freeboards, while ENVISAT provides thicker freeboards than CS2 in regions that are dominated by FYI.'*
*Give some figures; how much thinner and thicker.*
-Response:
We added a reference to the respective figure from the Paul et al., 2018 paper so that the reader can find the minimized differences between ENVISAT and CS2 freeboard. We also mention the average difference in the 'Data limitations' subsection (see Line 197) that has been moved from the 'Discussion' to the 'Data and methods' section.
-Changes:
Added reference to the figure in Paul et al., 2018 (**LINES 140**).

- *(11) 127: 'a weighted mean sea ice draft value.' What is this weighting?*
  -Response:
  Since multiple satellite data points fall into the 25 km radius around the moorings we did not just calculate an average over all of these values for the comparison to the VAL data but also considered the distances between the individual satellite data points to the mooring location. Closest satellite data points account for a larger fraction of the mean than data points that are further away. The fraction with which each data point is adding to the mean is dependent on its distance to the mooring.
  -Changes:
  Reviewer#3 had a very similar comment and we clarified the weighted mean in the text (**LINES 161-171**).

- *(12) Section 2.2.2: Give typical uncertainty in a single SIT orbit value. How many SIT points are typically averaged within 25 km radius and in a daily scale? What is the uncertainty of this average?*
  -Response:
  As mentioned before, due to the variability in overflights between the months and dependent on the exact path relative to and overlap with the 25 km area around the moorings the number of data points that are averaged for the comparison to VAL data is very different. With the two different approaches (satellite versus mooring-based point measurements) we have to accept the fact that we are not going to be able to measure the exact same thing. Satellite data uncertainties are high for the measurements as well as for the parameters that go into the processing. The selected approach to utilize all available data within the vicinity and calculate a weighted mean is a measure to achieve best possible comparability under the given circumstances. We looked at the numbers of values that go into the weighted mean of one orbit trajectory (over the 25 km area around the mooring) and found numbers between 30 and 60 data points. But as mentioned before these numbers

can be very different from case to case and we made sure to account for these differences by weighting whatever values go into the average depending on how far away they are from the exact mooring location. The same holds for the uncertainty of the averaged values, they depend on the uncertainty of the individual orbit data points and the number and the variance of the values that go into the average themselves. They are very different for each individual data point but due to the noise and variance likely higher than the typical uncertainties of the single SIT orbit values.
-Changes:
We added the typical uncertainty of one SIT orbit value for ENVISATorbit and CS2orbit to the corresponding section in 'Data and methods' (ENVISATorbit: approx. 1.5 m, CS2orbit: approx. 1.1 m, **LINE 146**).

- *(13) Comment correlations shown in Figure 2 in Section 3.1.*
  -Response:
  We are not sure what correlations you are referring to here. We are not showing any correlations in Fig. 2 and therefore do not mention any in the text.
  -Changes:
  No changes.

- *(14) Did you investigate SIT anomalies in different part of the Laptev Sea? I think AARI uses Eastern and Western Laptev Sea regions. Is it possible to compare the SIT anomaly trends to any other study/data source? E.g. based on AARI ice charts? Are the trends related to polynya activity (extent, ice production) in the Laptev Sea?*
  -Response:
  We decided to investigate SIT anomalies for the entire Laptev Sea (as defined in Fig.1) and not divide this into Eastern and Western Laptev Sea mainly due to the fact that our available sonar data originates from moorings that are scattered all over the Laptev Sea. It is certainly possible to compare the SIT anomaly trends to other data sources, however, the focus of this study was on the comparison to high resolution sonar measurements, also for the investigation into stability of the CCI-2 sea ice data. We presented the Laptev Sea SIT anomaly as the basis for our investigation and provide a conclusion on whether the changes are based on satellite performance and how we interpret satellite SIT data in this region. However, as we mention in the 'Conclusion' the presented satellite SIT anomaly needs to be further investigate to understand the observed trends and the reasons behind them. This is beyond the scope of this study and will be tackled in future studies.
  -Changes:
  No changes.

- *(15) Figure 3(b) is not commented/discussed in the text; e.g. symmetry/normality of the pdf?*
  -Response:
  Figure 3b is now described in the text.
  -Changes:
  Short description of Fig. 3b in **LINE 220**.

- *(16) Section 3.2 title could be 'Gridded monthly sea ice draft'.*
  *Section 3.3 title 'Higher temporal resolution satellite products' is not good, what is this 'higher'?*
  *'Daily and weekly sea ice draft products'?*
  -Response:
  Apparently, you are not the only one who thought that these titles should be different. Thank you for commenting on this, we followed suggestions from Reviewer #3 to slightly change the structure of this section and accordingly update the titles. We used your suggestion 'Gridded monthly sea ice draft' in the process. We hope you agree with the changes that we made.
  -Changes:
  Changes were made to the structuring of the 'Results' section, including new titles.

- *(17) In Figure 4 it is difficult to see grey crosses. Maybe these single data points can be removed and instead describe in the text how many RA draft data points were typically in each VAL draft interval.*
  -Response:
  Thank you for bringing this up. We think that it is really important to not just show the binned averages but also the data points that combine for those values. However, since the number of data points per bin is very variable it does not really make sense to mention these numbers in the text, we feel that this is better covered by showing the raw data in the figure. In order for the reader to really see those crosses we changed the color to black. We hope this improves readability of the figure.
  -Changes:
  Figure 4 colouring of the raw data (crosses) was changed from grey to black.

- *(18) 214: 'It also confirms the intermission biases between ENVISAT and CS2 that were published by Paul et al. (2018).' Please give out these biases here.*
  -Response:
  Following a comment from Reviewer #2 we moved the 'Data limitations' section from the 'Discussion' to 'Data and methods', there we specifiy the intermission bias for the Laptev Sea. We also added the reference to the respective figure in Paul et al., 2018 to the 'ESA CCI-2 monthly mean gridded

product' section ('Data and methods', your specific comment #10). There-
fore, intermission biases have been mentioned and specified leading up to this
comment and we think it is not necessary to mention them again here. We
hope you agree.
-Changes:
Changes to the general structure were made to account for this comment.
No changes were made to this specific sentence though.

- *(19) 227: 'Consequently the underestimation of sea ice draft with increas-
  ing thickness is largest for CS2SMOS because of the larger uncertainties of
  SMOS over thicker sea ice.' But for thicker ice CS2MOS SIT comes only
  from CS2 data? If so then SMOS uncertainty should have no effect here.*
  -Response:
  First of all, you are right the uncertainty of SMOS should not be the rea-
  son for the larger underestimation of VAL sea ice draft from CS2SMOS.
  CS2SMOS is based on an optimal interpolation, that means that both data
  products 'contribute' to the final value. This contribution is dependent on
  the uncertainty of the individual data points over the area in question. The
  underestimation observed for the CS2SMOS product is likely a result of local
  thin ice patches in the region that lead to a larger contribution of SMOS data
  to the final interpolated merged CS2SMOS SIT value.
  -Changes:
  We cut 'uncertainty' from the sentence in question and clarified that the un-
  derestimation of the CS2SMOS product is based on the influence of SMOS
  data on the final interpolated SIT product and not its uncertainty (**LINE
  271-272**).

- *(20) Figure 6(b) is not discussed in the text.*
  -Response:
  We discuss Fig. 6 as a whole in the text and feel that this is enough here.
  -Changes:
  However, we changed the histogram plot in Fig. 6(b). Rather than showing
  the same PDF as in Fig. 3 we now distinguish between the distributions of
  the selected thickness ranges (same as in panel (a) of Fig. 6). We accordingly
  updated the figure caption here.

- *(21) What data is ICETrack using? SAR imagery? Describe in the text.*
  -Response:
  A short summary of the motion products that are used by ICETrack has
  been added to the Taymyr case study section.
  -Changes:
  **LINES 341-346**

- (22) 346: 'That means that, for investigations into the sea ice cover in the Laptev Sea it is important to be aware that sea ice can persist some time after the presented satellites stop providing SIT data.'

  How about before the winter in late summer? Phrasing here: satellites do not provide SIT data, but your data processing methods do, i.e. SIT estimation is not possible (at least currently) for summer/melting season.

  -Response:

  The sentence you are referring to has been changed accordingly.

  -Changes:

  **LINES 370-372**

- (23) 350: 'The ESAs CCI-2 gridded SIT CDR covers a period from 2002 to 2017 and has been validated for multiple regions around the Polar regions of the Earth.'

  A short summary about the results of these validation activities would be good here. Later in Conclusions you could summarize what new insight in the accuracy of the CCI-2 SIT CDR your study resulted.

  -Response:

  Although this study is mainly focused on the validation of satellite SIT data in the Laptev Sea we agree that a short comparison and summary is beneficial to the 'Conclusion' section of this study.

  -Changes:

  Additional sentences on previous efforts (**LINES 374-376**) and the comparison to our results (**LINES 391-392**) were added to the text.

- (24) 378: 'Therefore, improvements in the processing of radar altimetry data are required for the estimation of surface roughness but also for the parametrizations of snow depth and densities of snow and ice.'

  How surface roughness would be utilized in the freeboard tracking? How about different freeboard trackers for different ice types, like FYI and MYI?

  -Response:

  Surface roughness widens the leading edge of the radar waveform and this information is used in the Envisat retracker to define the retracking point (Paul et al., 2018). A similar algorithm for CryoSat-2 synthetic aperture waveforms is currently under development and that is what our statement of needed improvements referred to. As for the question about different retracker algorithms for different ice types, this information has to be known on a per-waveform basis. This is currently not the case.

  -Changes:

  No changes.

- (25) 383: 'Furthermore would continuous long-term SIT measurements in the Laptev Sea provide much needed information on deformation processes.'

*Is this supposed to be an interrogative clause? Or 'Furthermore, continuous long-term SIT measurements in the Laptev Sea would provide.'?*

-Response:

-Changes:

Sentence has been revised (**LINE 410-411**).

**Additional changes from the authors**

- *(1) Due to changes in the review process of the ADCP sea ice draft derivation method paper (previously Belter et al., 2019b, now Belter et al., 2020b, in review at the Journal of Atmospheric and Oceanic Technology) the estimated uncertainty values provided for the daily mean sea ice draft time series have been changed. See changes in* **LINE 116-117 and LINE 179-181**.

- *(2) Daily mean sea ice draft time series from the Laptev Sea ADCPs have been published and a reference was added to the 'Data availability' section* **(LINE 415)**.

Finally, we would like to thank you again for your efforts to help us improve our manuscript.
Kind regards,
H. Jakob Belter

---

## Author Comment (AC2) · 16 Mar 2020

Point-by-point response to editor and reviewer concerns by
corresponding author: H. Jakob Belter
March 16, 2020

**tc-2019-307:**

**Satellite-based sea ice thickness changes in the Laptev Sea from 2002 to 2017: Comparison to mooring observations**

https://doi.org/10.5194/tc-2019-307:

Belter, H. Jakob, Thomas Krumpen, Stefan Hendricks, Jens A. Hoelemann, Markus A. Janout, Robert Ricker, Christian Haas

Dear anonymous Reviewer #2,

on behalf of all authors, I would like to thank you for this extremely helpful review. Below, we provide you with a point-by-point response to your comments. We hope that we were able to answer all your comments sufficiently. We would also like to refer you to the responses to the other reviewers for more improvements and changes to the manuscript.

General comments:

- *(1) The 'Data and method' section lacks important details. The data section only briefly introduces different data sets. It is not clear how many measurements were compared and how the mean Laptev Sea sea ice thickness from gridded data was calculated. Section 4.1. and 4.2 describing data limitations can be moved to the 'Data and methods' to provide the reader with valuable information before introducing the results.*

  -Response:

  We added more information, especially on the processing of ADCP sea ice draft to the respective subsection in the 'Data and methods' section. We also added an additional paragraph to the 'Sonar-based ice draft measurements' subsection to clarify how weekly and monthly VAL draft values have been calculated. Further changes have been made to the 'ESA CCI-2 orbit data' subsection to indicate how many data points are used for the comparison to VAL data (also in response to Reviewer #3). Also in response to the other reviewers we combined the averaging of the satellite SIT values into a single paragraph at the end of the 'Satellite data' subsection. The explanation of how ESA CCI-2 Laptev Sea SIT anomaly was calculated is provided in the first paragraph of the 'Results' section. Thanks to your great suggestion to move the 'Data limitations' subsection to the 'Data and methods' section we also provide additional information before presenting the results, which we agree provides the reader with more information to better follow the rest of this study. We hope you agree that the 'Data and methods' section is more comprehensive now.

  -Changes:

  Changes to the 'Sonar-based ice draft measurements' subsection (**LINES 121-124**). Additional changes to the 'ESA CCI-2 orbit data' subsection (**LINES 146-150**). New paragraph to combine satellite sea ice draft averaging at the end of 'Satellite data' subsection (**LINES 162-171**). 'Data limitations' subsection was moved to the end of 'Data and methods' section (**LINES 172-200**).

- *(2) The discussion can be elaborated. - The Anabar, Khatana and Lena stations a located in the area of polynya formation. Are polynya events taken into account in SAT and VAL data? Do the polynya events affect your comparison between SAT and VAL monthly mean? - One of the main finding shows that SAT data represents modal sea ice thickness rather than mean. What is a possible explanation? - The SAT-VAL difference depend on sea ice thickness. It there a seasonal change in this difference? I suggest that a scatter plot with seasonal cycle might be informative. - Section 4.4 introduces new data, method and results. Would it make more sense to restructure it*

*and add a subsection to methods and results? Why other data from ADCP and ULS is not shown? Does it confirm you findings?*

-Response:

Since you are asking a whole bunch of questions here we will divide this response into subtopics:

Polynya influence:

First of all, you are right that Anabar, Khatanga and Lena stations are in the area of possible polynya formation, however, the impact of polynya events on the VAL data is rather limited. In cases of open water (in the polynya) both ADCP and ULS are able to identify the lack of ice. In cases where thin ice is present in the polynya the instruments recognise the ice as well. Daily, weekly and monthly averages of sea ice draft are calculated after open water was exclude from the draft time series and therefore do not impact the final daily, weekly or monthly values. However, daily, weekly and monthly averages of sea ice draft are only calculated in cases where 'enough' data points are available. If for example 20 out of 30 days in October show open water (or no data for that matter), no October mean value is calculated for the respective mooring. The threshold for calculating any of the averages is 50% of the maximum number of data points that are available for that period. If more than 50% of the data is missing or attributed as open water no average is calculated. As for the SAT data, open water is also not included into the SIT values. Thin ice is a little more complicated as it is an issue of ongoing research. The problem here is whether the satellite detects thin ice as ice or as water. Is it detected as water, it does not influence the final SIT value. Thin ice on the other hand is very much overestimated just because of the fact that the algorithm predefines the same amount of snow it would add to thicker ice. For the presented comparison these difficulties from the SAT data side should not be an issue since our VAL data defines whether data points are compared or not. If the sonar detects a long period of polynya induced open water, no daily, weekly or monthly value is calculated. Accordingly there is no VAL data point that could be compared to the SAT data.

Explanation for better agreement with modal sea ice draft:

This issue is also actively discussed right now. The first step to identify reasons for this result would be to figure out whether the initial freeboard measurements already show this tendency. However, the available VAL data is based on draft measurements and no additional information on freeboard is available for comparison here. Possible reasons for this bias are mentioned in this study and include: errors in the retracking, surface type classifications, snow depth, ice density. It is very likely that a combination of all these factors contribute to the overall bias. In order to quantify them future comparisons with ICESat-2 data are planned.

Seasonal changes:

Considering that SIT has a seasonal cycle and SAT-VAL difference is dependent on thickness the SAT-VAL difference will definitely have a seasonal cycle as well. Thick winter ice is underestimated by the satellites (most negative SAT-VAL difference of the respective time series), while thin ice is overestimated (most positive SAT-VAL difference of the respective time series). It would definitely be interesting to look at seasonal changes especially for long-term data sets from one location, however, our data is limited to one year time series from all over the Laptev Sea and this is not possible here. Furthermore do we feel that this is not relevant for the study as it is. The agreement is thickness dependent, independent of when thick or thin ice is observed. The aim, especially of the 'Stability' subsection is to investigate the performance of the satellite data. Answering questions like: Is the SAT-VAL difference the same when 1 m thick ice is measured in 2003 compared to when 1 m thick ice is measured in 2015?

Taymyr 2013/2014 case study:

We thoroughly discussed whether the case study should be part of 'Data and methods' and 'Results', however, we came to the conclusion that it is more of an addition that highlights the results presented before. The introduction of the new method (using ICETrack to calculate accumulated convergence) is an add on here in order to give a first explanation about what possible reasons for the underestimation of thicker ice by the satellites could be. Something similar was commented by Reviewer #1. The analysis on whether the agreement between modal or mean VAL draft is better with satellite draft data was done only for ULS-based draft time series. The temporal resolution of the ADCP-derived drafts was simply not appropriate to calculate meaningful modal ADCP draft values. The result that satellite sea ice drafts agree better with modal VAL drafts was confirmed for all four ULS data sets. We only chose the Taymyr example here since it was the only one of the four ULS data sets that showed a gradual increase in mean sea ice draft (Taymyr 2013/2014 Jan to March). Accordingly the difference between mean and modal VAL draft is larger and our finding could be visualized best.

-Changes:

Polynya influence:

We added the information that open water values were excluded prior to averaging of VAL draft data (**LINES 121-124**).

Taymyr 2013/2014 case study:

We added a sentence on why the comparison to modal values was only done with ULS data (**LINE 364-366**).

Specific comments:

- *(1) Line 160: 'The ESA CCI-2 SIT CDR shows an overall thinning of sea ice in the Laptev Sea between 2002 and 2017.' The sentence about is too strong. The error of the overall trend is as large as trend. Also the significance of the trend is quite low. The black line rather shows that there is no changes in sea ice thickness.*
  -Response:
  Thank you for pointing that out. You are right these first few sentences concerning the overall trend are too strong.
  -Changes:
  We toned down the sentences in question to make sure the reader realizes that although the trend line is slightly negative it is highly uncertain and should rather be interpreted as no significant trend over the period from 2002 to 2017 (**LINES 203**).

- *(2) Lines 161-162: How is the Laptev Sea defined? Please show the region used for SIT anomaly calculation in Figure 1.*
  -Response:
  -Changes:
  The region used to calculate ESA CCI-2 SIT anomaly was added to Fig. 1. Fig. 1 and Fig. 2 captions were updated.

- *(3) Line 210: 'ENVISATorbit data shows a higher average RMSD, stronger average underestimation of VAL sea ice draft and much lower average correlation with VAL sea ice drafts compared to the gridded ENVISAT data' Is there an explanation? Why does the orbit data which supposed to be closer to the VAL measurement shows worse statistical characteristics?*
  -Response:
  This is very likely related to the uncertainty of the individual orbit values and the larger number of data points that go into the weighted mean and the corresponding larger noise compared to the gridded data sets.
  -Changes:
  Also in response to a comment from Reviewer #1 the uncertainties of ENVISAT and CS2 orbit SIT values were added to the respective 'Data and methods' subsection (**LINE 146**).

- *(4) Lines 282-283: 'The seasonal biases between ENVISAT and CS2 need to be considered for the temporal development of the Laptev Sea SAT-VAL differences between the two periods'. Please elaborate. Are those biases considered in this study?*
  -Response:

What that means is that simultaneously measured SIT values from ENVISAT and CS2 are different from one another. This seasonal bias is strongly connected to the different ice thicknesses that can be observed over the course of a 'season' (which is basically late-autumn, winter and early-spring, since CCI-2 SIT data is only available from October through April). These differences are introduced in the 'Data and methods' section (2.2.1). This sentence here serves as a reminder that these inter-mission biases exist and that they are not constant throughout a single season. We consider these biases here, as we compare monthly mean values of sea ice draft from ENVISAT and CS2 rather than annual averages. The offsets between ENVISAT and CS2 to VAL data are shown in Fig. 3, 4 and 5. However they are not declared or displayed as seasonal but thickness dependent offsets. Biases between ENVISAT and CS2 can be specifically seen in Fig. 3 where ENVISAT-VAL and CS2-VAL differences are plotted for the Outer Shelf stations (2010/2011 and 2011/2012). As the focus of this study is on whether satellite products are stable over time we are concerned with thickness rather than seasonal values. We do not want to show whether ENVISAT and CS2 show the same agreement or offset every March but whether they show constant biases for, for example, 1 m thick ice independent of when 1 m thick ice occurs.
-Changes:
No changes.

- *(5) Lines 343-348: It is worth mentioning that ULS provide sea ice draft measurements after the onset of melt. However it is not a real finding that there is sea ice in the Laptev Sea in June-July. Please consider reformulating.*
-Response:
We apologize if it seemed like we presented this fact as a new finding. We were merely trying to remind the reader that the temporal limitations of the radar altimeter satellite products should not be mistaken for complete loss of sea ice after April.
-Changes:
We reformulated the above-mentioned paragraph (**LINES 367-372**).

- *(5) Line 359: 'The presented satellite products represent similar sea ice drafts differently.' I am not sure the meaning is clear. Do you mean identical sea ice draft or sea ice draft of similar thickness, e.g. within presented bins?*
-Response:
Thank you for bringing this up. We are referring to Fig. 5 where it is indicated that the same VAL sea ice draft values are represented very differently by the five investigated SAT data products.
-Changes:
We clarified that in the 'Results' section (**LINES 280-281**).

Technical comments:

- *(1) Page 1 line 24: sea ice system - sea ice state?*
  -Response:
  -Changes:
  Changed.

- *(2) Page 2 Line 43: a space after '(ULS)' is missing.*
  -Response:
  -Changes:
  Corrected.

- *(3) Line 132: a space after 'ENVISAT' is missing.*
  -Response:
  In this case we are using the abbreviation ENVISATorbit that was introduced in the line above.
  -Changes:
  No changes required

- *(4) Figure 2: It seems that colors of the legend in the upper left corner are mixed up. The negative trend should be the ENVISAT one.*
  -Response:
  -Changes:
  Corrected.

- *(5) Figure 7: The scale on the sea ice draft axis is missing.*
  -Response:
  We are not sure what you meant with the missing scale. The sea ice draft axis (left) shows sea ice draft values from -1 to 5 m and is labelled with the corresponding unit (m). It is the reference axis for mean, modal, CS2 and CS2SMOS sea ice draft, while the axis to the right corresponds to accumulated convergence from the NSIDC (both the axis and the acc. convergence graph are given in blue).
  -Changes:
  No changes.

**Additional changes from the authors**

- *(1) Due to changes in the review process of the ADCP sea ice draft derivation method paper (previously Belter et al., 2019b, now Belter et al., 2020b, in review at the Journal of Atmospheric and Oceanic Technology) the estimated uncertainty values provided for the daily mean sea ice draft time series have been changed. See changes in* **LINE 116-117 and LINE 179-181**.

- *(2) Daily mean sea ice draft time series from the Laptev Sea ADCPs have been published and a reference was added to the 'Data availability' section* **(LINE 415)**.

Finally, we would like to thank you again for your comments and great suggestions. We hope you agree that the changes made improve the manuscript.
Kind regards,
H. Jakob Belter

---

## Author Comment (AC3) · 16 Mar 2020

Point-by-point response to editor and reviewer concerns by
corresponding author: H. Jakob Belter
March 16, 2020

**tc-2019-307:**

**Satellite-based sea ice thickness changes in the Laptev Sea from 2002 to 2017: Comparison to mooring observations**

https://doi.org/10.5194/tc-2019-307:

Belter, H. Jakob, Thomas Krumpen, Stefan Hendricks, Jens A. Hoelemann, Markus A. Janout, Robert Ricker, Christian Haas

Dear anonymous Reviewer #3,

on behalf of all authors, I would like to thank you for your comments and suggestions to our manuscript. Please find our point-by-point response to your review below. We hope you agree with our changes and feel that your comments have been answered properly. We would also like to refer you to the responses to the other reviewers for more improvements and changes to the manuscript

Specific comments:

- *(1) Line 12: This phrase does not fully correspond to the results presented in the paper. Overestimation (underestimation) of sea ice draft for thin ice below 0.7 m (thick ice above 1.3 m) is indicated from comparison of the mean values, but not with respect to the modal draft.*
  -Response:
  You are right, this sentence was a little misleading we revised it accordingly.
  -Changes:
  **LINES 12-14**

- *(2) Line 40: Authors could also note that in (Kern et al., 2018) the airborne Operational Ice Bridge data were used for validation of the satellite product as well.*
  -Response:
  Thank you, we added Operation IceBridge to the list of observational data sets that have been used for the validation of the CCI-2 SIT products.
  -Changes:
  **LINES 43-44**

- *(3) Section 2.1: I guess that open water was excluded from the sonar-based measurements. If so, please, mention it in the text.*
  -Response:
  We added a paragraph on the VAL data averaging and a sentence clarifying that open water values were not included in the calculations of averages.
  -Changes:
  **LINES 121-124**

- *(4) Line 101: The phrase 'bottom track mode measurements of surface and error velocity' sounds not clear. Although paper by Belter et al. (2019b) will, I guess, describe details of the methodology, some clarifications on what is, e.g., 'error velocity' would be helpful.*
  -Response:
  You are quite right that this description is not sufficient. We therefore added a couple of explaining sentences and the reference to the ADCP sea ice draft data on the PANGAEA data archive where a short summary of the processing steps is provided.
  -Changes:
  Also in response to Reviewer #1, did we add further explanations to the respective section (**LINES 112-116**). Furthermore, did we publish a short summary of the processing steps with the ADCP sea ice draft data sets on PANGAEA (https://doi.pangaea.de/10.1594/PANGAEA.912927).

- *(5) Line 125: This way of estimating draft uncertainty is applicable if SIT uncertainty accounts for the sources of freeboard uncertainty. If so, please, mention it in the text. From the other side, since the authors do not use this draft uncertainty further in the analysis, it is not clear what it was estimated for.*

  -Response:

  Thank you for bringing this up. As mentioned in the response to Reviewer#1 the presented draft uncertainty is more complex than what we presented. Following your final comment regarding this issue we removed the draft uncertainty estimation completely and also removed the uncertainty bars from Fig. 7. Since uncertainty is still a very important parameter for the analysis we added information about satellite SIT uncertainty in the respective 'Data and methods sections'.

  -Changes:

  Uncertainty equation and uncertainty bars in Fig. 7 have been removed. Information about uncertainties have been added to the respective 'Data and methods' subsections.

- *(6) Line 127: It is not clear how authors calculate weighted mean values. Possibly this weighting account for the distance between grid center and mooring location? If so, please clarify it.*

  -Response:

  You are completely right, weighted means account for the distances of the selected satellite data points to the mooring location.

  -Changes:

  We clarified that in the text (**LINES 162-171**).

- *(7) Line 133: As a frequency of the orbit tracks that pass over the mooring sites the authors specify 'four overflights'. However Envisat and Cryosat-2 have different orbit inclinations and this frequency should be different for these missions.*

  -Response:

  You are right the number of overflights is different for ENVISAT and CS2. In fact, the number of overflights is occasionally below and some times above four for individual months. We were not accurate enough in the commented sentence. We clarified that the number of overflights is different for the two satellites and also changed 'approximately' to 'about'. On average the number of overflights is even more than four for both satellites. However, the point of the sentence was not to provide a fixed number of overflights per month (which does not exist since it varies) but to indicate that in any case orbit data provides more data points for the comparison to VAL data than the monthly mean gridded CCI-2 data. We hope you agree with the changes

we made to clarify the sentence.
-Changes:
Sentence was changed (**LINES 146-150**).

- *(8) Section 3: I suggest the authors to change structure of this section: to combine sections 3.2 and 3.3 in one section 3.2 with the title, for example, 'Validation of CCI-2 products', and with the subsections '3.2.1 Gridded CCI-2 sea ice draft' (currently section 3.2), '3.2.2 Orbit CCI-2 sea ice draft' (currently section 3.3.1), and '3.3.3 Intercomparison of CCI-2 and merged CS2SMOS drafts' (currently section 3.3.2). Then the accordingly revised text from the first paragraph of the current section 3.3 could be moved to the beginning of new section 3.2*
-Response:
Thank you for this suggestion. Reviewer #1 suggested changes to the titles of this section and we agree that your suggested structure improves the readability of this section. However, we were a little confused by your suggestion to move the first paragraph of the old 3.3 section to the beginning of the new 3.2 section. Since 3.2 is the overarching section for the comparison between all satellite and VAL data we moved the paragraph in question to the end of the new 3.2.1 section. We feel that it fits here since we finish the results part of the monthly mean gridded data and show the reader that we are moving on to the higher temporal resolution products here. We hope you agree with this change. However, calling the last section 'Intercomparison...' is a really good suggestion since the CS2SMOS/VAL comparison is rather short and the paragraph focusses on the results of all presented satellite products.
-Changes:
Changes were made to the structuring of the 'Results' section, including new titles.

- *(9) Line 228: I guess that this enhanced underestimation of thick ice by CS2SMOS data is observed because for some bins corresponding to thick ice the CS2SMOS product is the only available data (as I can see from Figure 5). It means that for these bins CS2SMOS product is generated only from the SMOS measurements. If so, this could be explained in the text.*
-Response:
This is an interesting observation but you have to remember that Fig. 5 shows satellite data from products with different temporal resolutions. Gridded ENVISAT/CS2 data (filled circles) is based on the initial orbit data (empty circles) and CS2SMOS is based on the gridded CS2 and SMOS data. 'Missing' thicker CS2 data could also be caused by missing VAL (when there is no VAL data point available there is no comparison), for example due to a long open water periods that prevented the generation of a monthly mean VAL sea

ice draft value (see new paragraph on open water influence and calculation of VAL daily, weekly and monthly averages, end of section 2.1). However, we agree that reason for this increased underestimation is the influence of SMOS data on the merged product and this could very well be caused by a total lack of CS2 data, resulting in SMOS data being the only product defining an individual data point. In response to a comment from Reviewer #1 we revised the sentence about the SMOS influence to show that the larger impact of SMOS leads to the increased underestimation. We hope that this answers your comment as well.

-Changes:

Additional paragraph at the end of section 2.1 (**LINES 121-124**) and changes to the sentence in the 'Results' section (**LINES 271-272**).

- *(10) Line 285: The reference (Paul et al., 2018) is not appropriate here. Paul et al., 2018 do not provide regional estimates of the differences between SIT derived from ENVISAT and CS2 data.*

  -Response:

  You are right, Paul et al., 2018 provide maps showing the differences between ENVISAT and CS2 freeboard. The value we are presenting here has been calculated from the available SIT data sets.

  -Changes:

  Reference to Paul et al., 2018 was removed from the sentence (**LINE 197**). Please note that the entire 'Data limitations' subsection was moved to the 'Data and methods' section (following a comment from Reviewer #2).

- *(11) Line 315: The indicated trends are small, that supports the conclusion that the gridded CCI-2 CDR is stable over considered period. However Fig.6 shows that these trends might be caused not only by the intermission differences. The trends for thickness ranges 0 to 1 m and 1 to 2 m looks negative even separately for Envisat and CS2 data as well as for combined dataset. For thickness range 2 to 3 m two overlapping points in 2011 shows that Envisat rather overestimate sea ice draft as compared to CS2 as well as for thinner ice.*

  -Response:

  First of all, we agree with your assessment that these trends are small and support the conclusion that the gridded CCI-2 CDR is stable over the considered time period (we mention that in the text as well). You are also right that these trends seem to be there even for each of the two satellite periods individually, however, since the number of data points available for each thickness range is already very small over the full period, looking at both satellite periods separately would decrease this number even further and make those trends even more uncertain. Furthermore, should we consider that these values are not recorded at the same location, as we see from differences in the agreement between SAT and VAL data in other Arctic regions. This could be simply caused by regional differences in the performance of the satellite products. The reason why we attribute those trends for the three thickness ranges to the inter-mission bias is that fact that it agrees rather well with what Paul et al., (2018) found. The inter-mission bias seems to be dependent on thickness. ENVISAT overestimates (underestimates) thin (thick) ice compared to CS2. For the first two thickness ranges we would expect ENVISAT-VAL differences to be larger compared to CS2-VAL differences. This is the case for the thickness ranges from 0 to 1 and 1 to 2 m (as also suggested by the trend lines). However, we agree that the statement made for the 2 to 3 m thickness range might be a little strong. We revised the sentence to indicate that the small number of data points (even smaller than for the other two thickness ranges) and the underestimation by the satellites that strongly increases within the 2 to 3 m thickness range make this trend rather uncertain and we therefore call it inconclusive. We hope you agree with this change.
-Changes:
Revised sentence (**LINES 327-332**).

- *(12) Line 380: It can be noted here that not only snow depth, but specifically snow properties that influence the location of the main scattering horizon are a major source for uncertainty in the freeboard retrieval process.*
  -Response:
  Thank you for this important addition. We added snow properties to the sentence in question.
  -Changes:
  **LINE 407**

Technical comments:

- *(1) Line 79: I suggest to replace 'approaches' by 'instruments', otherwise one may interpret it that both ADCP and ULS data are processed by two methods.*
  -Response:
  -Changes:
  Corrected.

- *(2) Line 101: Abbreviation 'BT' is not needed here as it is not used further in the text.*
  -Response:

-Changes:
Corrected.

- *(3) Figure 2: Colours of the first and second lines indicating trend values should be switched*
  -Response:
  -Changes:
  Corrected.

- *(4) Line 231: I suggest to reformulate this sentence: 'While individual stations deviate from this average the overall tendency indicates a dependency of the agreement between monthly mean gridded CCI-2 and VAL sea ice draft on sea ice thickness.'*
  -Response:
  Thank you for pointing this sentence out. It is a little confusing.
  -Changes:
  We changed the entire sentence (**LINE 275-278**).

- *(5) Line 332: In the 'newly formed FYI ice' the word 'ice' is not needed.*
  -Response:
  -Changes:
  Corrected.

- *(6) Line 383: The sentence 'Furthermore...' sounds not clear. Please, consider revising.*
  -Response:
  -Changes:
  Sentence has been revised (**LINE 410-411**).

- *(7) Table 4: In the captions to the Table it is noted that the statistical parameters 'were calculated for the four VAL data sets'. However this table presents the results only for two stations with ULS measurements: Taymyr and 1893.*
  -Response:
  We are looking at Taymyr and 1893 data from the 2013/2014 and 2014/2015 periods which results in a total of four different data sets.
  -Changes:
  We clarified this in the Tab. 4 caption.

**Additional changes from the authors**

- *(1) Due to changes in the review process of the ADCP sea ice draft derivation method paper (previously Belter et al., 2019b, now Belter et al., 2020b, in review at the Journal of Atmospheric and Oceanic Technology) the estimated uncertainty values provided for the daily mean sea ice draft time series have been changed. See changes in **LINE 116-117 and LINE 179-181**.*

- *(2) Daily mean sea ice draft time series from the Laptev Sea ADCPs have been published and a reference was added to the 'Data availability' section (**LINE 415**).*

In the end we want to thank you again. We really appreciate your input and hope you agree that the manuscript has improved.
Kind regards,
H. Jakob Belter

---

## Referee Report (RR1)

**Satellite-based sea ice thickness changes in the Laptev Sea from 2002 to 2017: Comparison to mooring observations**

Hans Jakob Belter, Thomas Krumpen, Stefan Hendricks, Jens Hoelemann, Markus Janout, Robert Ricker, and Christian Haas

https://doi.org/10.5194/tc-2019-307

**25 May 2020**

The authors have presented detailed, proper answers to all my comments to the first version of the paper, and made corresponding changes and additions to the paper. I think that the paper has improved considerably, and its overall structure is now good. Below I have some minor comments for your consideration for further possible paper improvements.

2) In Introduction Section you could review what is current understanding on the accuracy/quality of the CCI-2 SIT CDR: it seems this has been investigated at least by Kern et al. 2018. Are there any other studies, especially in peer-reviewed journals? You could also review similar other studies: comparisons between RA SIT records and sonar draft data. What is the typical relationship(s) between sonar and RA drafts over MYI?

No changes in the 'Introduction' but a short additional review of previous results and a comparison between MY and FYI results in the 'Conclusion'.

This is good addition, but you could also shortly review studies conducted (if exist) with other RA SIT records, e.g. by UCL and NASA,

3) A short section describing typical sea ice conditions and typical progress of sea ice season in the Laptev Sea would be good addition to the paper. How much there can be MYI in the Laptev Sea? Can there be large areas of grounded landfast ice for which the used freeboard to SIT conversion is not valid, and thus, could have an effect on your results?

We added a short paragraph to introduce the general conditions of the Laptev Sea ice cover to the 'Introduction' (LINES 52-57).

"with water depths between 15 and 200 m very shallow" somewhat odd sentence; "and it is very shallow with water depths between 15 and 200 m"?

6) In Section 2 you could have a sub-Section which describes how different datasets are processed to match each other. Now this information is scattered in sub-Sections describing the datasets. Also include a Table which summarizes datasets: spatial and temporal resolutions, accuracies, etc.

Summary paragraph for satellite data processing following the introduction of the satellite data sets LINES 161-171).

This is very good addition. It could be under its own sub-section; now it is after "2.2.3 Merged CryoSat-2/SMOS data"

9) Section 3.3.2 Merged CS2SMOS sea ice draft contains also a summary of all results; this should be in its own sub-Section.

We added a free line after the Section 3.3.2. to show that the summary below is the summary for the entire 'Results' chapter rather than the 3.3.2. Section.

-Changes: Addition of extra line after LINE 272.

I guess you can do it like this, but maybe you could check with the editor if this is in line with the TC paper style.

11) Tables 1-3 show averages of statistical parameters from different mooring locations. I am not sure this is meaningful, what an average correlation coefficient really tells us here? I think better would be here to combine all datasets together and then calculate RMSD, mean difference and r.

We hope you agree that we clearly state that the averages of the correlation coefficients are nothing more than averages over all stations and do not provide information about how good the general agreement between VAL and the respective satellite data is. No additional changes have been made here.

Yes, I can agree on your solution.

It would be interesting to see what is the typical variation of the sonar draft during a day, week and month. A figure about a time series of sea ice draft from some ADCP location would be nice.

An example of the variation on a monthly scale is given in Fig. 7 (the Taymyr case study) and we are happy to provide the high frequency (1 Hz) sea ice draft time series (also for the Taymyr case) below (Fig. 1).

I think you could add this figure under Section 2.1.1 with short description in the text. It shows nicely much there is draft variation in the raw data vs. the daily mean. You could also add weekly and monthly averages to this figure. This would nicely show the ULS draft variation in different temporal scales.

Are there any peer-reviewed journal papers that could be put as reference to CCI-2 SIT CDR in Section 2.2.1? A figure about monthly gridded SIT over the Laptev Sea would be interesting see; also what it typical SIT spatial variation over the Laptev Sea in this monthly product? How many pixels there are over the Laptev Sea?

No changes to the text required.

I still think an example figure in Section 2.2.1 about monthly gridded CS2 SIT over the Laptev Sea would be interesting.

Comment correlations shown in Figure 2 in Section 3.1.

We are not sure what correlations you are referring to here. We are not showing any correlations in Fig. 2 and therefore do not mention any in the text.

There are correlations in Figure 2:

line 170: "Since all five data sets are based on radar altimetry data satellite sea ice draft data is only available from October through April."

Short explanation with references for why the temporal restriction (dry snow period). Later in lines 369-370 there is an explanation related to this.

The paragraph in lines 248-252 is somewhat separate from the topic of Section 3.2.1. This could be moved as introductory (with some intro also to 3.2.1) to beginning of Section 3.2.

---

## Author Response (AR2)

Response to the final reviewer comments
corresponding author: H. Jakob Belter
June 9, 2020

**tc-2019-307:**

**Satellite-based sea ice thickness changes in the Laptev Sea from 2002 to 2017: Comparison to mooring observations**

https://doi.org/10.5194/tc-2019-307:

Belter, H. Jakob, Thomas Krumpen, Stefan Hendricks, Jens A. Hoelemann, Markus A. Janout, Robert Ricker, Christian Haas

Following the final reviewer (Reviewer#1 and #3) comments we decided to add two additional subsections to the 'Data and methods' section. (**LINES 121 and 162**).

Reviewer#1:

- reply to 2):
  We hope that our comparison to previous studies (see 'Conclusion') on RA and sonar drafts is sufficient at this stage.

- reply to 3):
  The sentence was changed following your suggestion (**LINE 54**).

- reply to 6):
  See reply above.

- reply to 9):
  Thank you for this comment. We will check with the Publications Production Office.

- reply to 11):
  Thank you!

- rely to figure request:
  We still feel that an additional figure would lengthen the manuscript unnecessarily. However, we are currently working on another publication that focuses on the variability of SIT in the Laptev Sea. We will of course show variations on various time scales there.

- reply to Correlations comment:
  Thank you for clearing this up. Actually, $R^2$ is not a correlation but the 'Goodness-of-Fit' of the trend line to the data points. We hope that answers your question about correlations. We added a sentence to the caption of Fig. 2 to clear it up.

- reply to gridded monthly mean SIT figure:
  You are right this is an interesting thing to look at, however, since we are looking at all months rather than just a single one, adding those figures would significantly lengthen the manuscript. These plots for all months considered here are available online at (https://data.meereisportal.de/gallery/).

- reply to: *Short explanation with references for why the temporal restriction (dry snow period). Later in lines 369-370 there is an explanation related to this.*:
  Rather than adding another paragraph on this earlier we hope that one explanation of this is sufficient for the manuscript.

- reply to: *The paragraph in lines 248-252 is somewhat separate from the topic of Section 3.2.1. This could be moved as introductory (with some intro also to 3.2.1) to beginning of Section 3.2.*:
  We checked this section again and prefer the paragraph to remain at this position. After introducing the results from gridded CCI data we sort of look out to the following sections.

Reviewer#3:

- 1. See comment and changes above.

- 2. Thank you for this comment. We checked it again and it is correct this way.

On behalf of all authors, thank you again to all reviewers for your comments and suggestions. Without your input our manuscript would not have been accepted for publication. We really appreciate all the work you have put into this.
Kind regards,
H. Jakob Belter

[revised manuscript text omitted]